# Learning to Guide Human Decision Makers with Vision-Language Models

## Abstract

There is growing interest in AI systems that support human decision-making in *high-stakes* domains (e.g., medical diagnosis) to improve decision quality and reduce cognitive load. Mainstream approaches pair human experts with a machine-learning model, offloading low-risk decisions to the model so that experts can focus on cases that require their judgment. This ***separation of responsibilities*** setup, however, is inadequate for high-stakes scenarios. The expert may end up over-relying on the machine's decisions due to *anchoring bias*, thus losing the human oversight that is increasingly being required by regulatory agencies to ensure trustworthy AI. On the other hand, the expert is left entirely unassisted on the (typically hardest) decisions on which the model abstained. As a remedy, we introduce ***learning to guide*** (LTG), an alternative framework in which – rather than taking control from the human expert – the machine provides *guidance* useful for decision making, and the human is entirely responsible for coming up with a decision. In order to ensure guidance is *interpretable* and *task-specific*, we develop SLOG, an approach for turning *any* vision-language model into a capable generator of textual guidance by leveraging a modicum of human feedback. Our empirical evaluation highlights the promise of SLOG on both on a synthetic dataset and a challenging, real-world medical diagnosis task.

## 1 Introduction

High-stakes applications in healthcare, criminal justice and policy making can substantially benefit from the introduction of AI technology, yet full automation in these scenarios is not desirable, due to ethical, safety and legal concerns, if not explicitly forbidden by law (Government of Canada, 2019; European Commission, 2021). For these reasons, human-AI or ***Hybrid Decision Making*** (HDM) is becoming increasingly popular to tackle high-stakes tasks. HDM algorithms pair a human decision maker with an AI agent – often a machine learning model – capable of providing support, with the goals of improving *decision quality* and lowering *cognitive effort.*

Most current approaches to HDM follow a principle of ***separation of responsibilities***, in the sense that they route novel inputs to exactly one of the two agents – *either* the human *or* the AI – who is then responsible for coming up with a decision. Specifically, in existing approaches (Madras et al., 2018; Mozannar & Sontag, 2020; Keswani et al., 2022; Verma & Nalisnick, 2022; Liu et al., 2022; Wilder et al., 2021; De et al., 2020; Raghu et al., 2019; Okati et al., 2021), the AI first assesses whether an input can be handled in autonomy – e.g., it is low-risk or can be addressed with confidence – and defers to a human partner otherwise. These algorithms help humans focus on the cases the model flags as most needing attention.

We argue that this setup is *suboptimal* and potentially *unsafe*. It is suboptimal because, whenever the machine opts for deferral, the human is left resolving hard cases completely unassisted (as in Fig. 1, right). At the same time, it is unsafe, because humans are affected by *anchoring bias* (Rastogi et al., 2022; Eigner & Händler, 2024), a phenomenon whereby decision makers tend to blindly rely on an initial impression (the anchor) and refrain from exploring alternative hypotheses. When the anchor is provided by an algorithm, the bias is amplified as humans tend to over-trust the machine's decisions when available and ignore their own opinions, a phenomenon called *automation bias* (Cummings, 2012) (Fig. 1, middle). This effectively

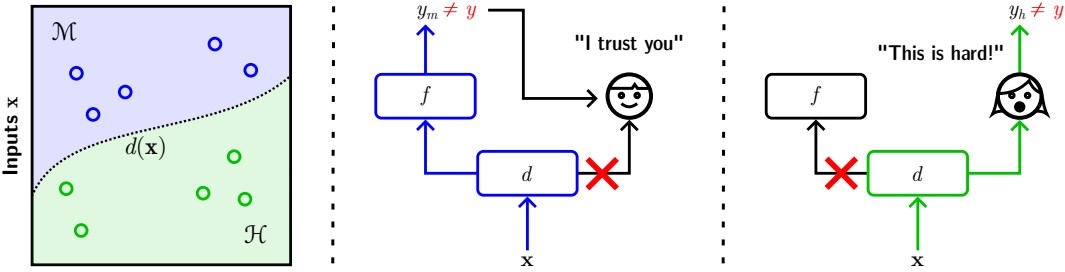

Figure 1: **Left**: Existing HDM algorithms employ a deferral function $d(\mathbf{x})$ to *partition* the input space $\mathcal{X}$ into $\mathcal{H}$ and $\mathcal{M}$. **Middle**: A predictor $f(\mathbf{x})$ handles the inputs falling in $\mathcal{M}$ (in **blue**). Because of *anchoring bias*, the human expert may end up blindly trusting its (possibly poor) decisions $y_m$. **Right**: The human, on the other hand, is left completely unassisted for those (possibly hard) decisions falling in $\mathcal{H}$, increasing the chance of mistakes in the human's decisions $y_h$ (in **green**).

undermines *human oversight* over algorithmic decisions, which is increasingly being required by governments around the world to regulate the use of AI in high-stakes applications (Green, 2022).

As a remedy, we propose **learning to guide** (LTG), an alternative setup that side-steps these issues. In LTG, the machine is trained to supply its human partner with interpretable *guidance* highlighting those aspects of the input that are useful for coming up with a high-quality decision. For instance, in pathology prediction, the guidance highlights the pathologies present in an input X-ray scan that are indicative of possible diagnoses. In LTG, *by construction*, all decisions are taken by the human expert – thus preventing automation bias – but facilitated by accompanying machine guidance.

We showcase LTG on **medical decision making** focusing on guidance formulated in **natural language**. Along with that, we also validated the effectiveness of LTG with a synthetic dataset. To this end, we introduce SLOG (Surrogate-based Learning to Guide), an algorithm for turning large vision-language models (VLMs) (Radford et al., 2021; Yan & Pei, 2022; Sharma et al., 2021) into high-quality guidance generators. In a nutshell, SLOG takes a VLM pre-trained for caption generation and fine-tunes it using feedback about the quality of downstream human decisions inferred from generated guidance. SLOG keeps annotation costs under control by training a *surrogate model* that predicts downstream decision quality on a modest amount of feedback, and then using it to fine-tune the VLM in an end-to-end fashion. Our experiments on a challenging medical diagnosis task indicate that VLMs fine-tuned with SLOG output interpretable task-specific guidance that can be used to infer high-quality decisions.

**Contributions.** In summary, we:

- Expose critical limitations in prevailing HDM algorithms that undermine their suitability for high-stakes decision-making.

- Propose *learning to guide* (LTG), a novel approach for assisting human decision-makers that keeps humans continuously in the loop.

- Present SLOG, an LTG approach tailored for natural language guidance that can convert large VLMs into interpretable, task-specific guidance generators.

- Demonstrate the effectiveness of SLOG on a challenging medical diagnosis task.

## 2   Hybrid Decision Making

We target decisions that must retain human oversight (e.g., medical diagnosis) because full automation poses safety risks.[1]   Research on HDM develops AI assistants to augment human experts on such tasks.

---

[1]We focus on classification problems, with inputs $\mathbf{x} \in \mathbb{R}^d$ and categorical or multi-label decisions $y$. Despite this, our remarks apply to other prediction problems as well, e.g., regression (De et al., 2020).

Considering the AI assistant and the human expert have different abilities, expertise, and biases, the central question of HDM is how to best integrate them.

**HDM by Separation of Responsibilities**. Existing HDM strategies solve this problem by following a principle of *separation of responsibilities*: any given instance $\mathbf{x}$ is assigned to exactly one of the two agents, who is then in charge of decision making, cf. Fig. 1. Specifically, they implement a *classifier* $f : \mathbf{x} \mapsto \hat{y}$, playing the role of an AI agent, as well as a *deferral policy* $d : \mathbb{R}^d \to \{\texttt{machine}, \texttt{human}\}$ that partitions the input domain $\mathcal{X}$ into two disjoint subsets, $\mathcal{M}$ and $\mathcal{H}$. Novel inputs $\mathbf{x}$ falling in the former are handled by $f$ and those falling in the latter are handled by the human expert. This setup is known under a variety of names, including *learning to defer* (Madras et al., 2018; Mozannar & Sontag, 2020), *learning under algorithmic triage* (Raghu et al., 2019; Okati et al., 2021), *learning under human assistance* (De et al., 2020; 2021), and *learning to complement* (Wilder et al., 2021; Bansal et al., 2021).

Approaches differ in how they partition the input space $\mathcal{X}$. Earlier methods build on *prediction with a reject option* (Cortes et al., 2016), in which the deferral policy $d$ observes all incoming instances $\mathbf{x}$ and offloads those about which the predictor $f$ is unsure (based on, e.g., predictive variance) (Raghu et al., 2019). Since $f$ is fixed, the partition is static and depends only on the self-assessed uncertainty of the predictor. Assuming the latter is sufficiently well calibrated (Kendall & Gal, 2017), this strategy can perform well in practice (Liu et al., 2022). The main drawback with this setup is that the partitioning accounts for the machine's performance only, neglecting the human's expertise and biases. Madras et al. (2018) improve on this by *learning* the deferral policy $d$ so that it optimizes some decision theoretic measure of *joint team performance*, thus explicitly taking the quality of human decisions into account. Follow-up works (De et al., 2020; 2021; Wilder et al., 2021) go one step further and train the deferral policy $d$ and the predictor $f$ *jointly*, so as to adapt one to the other. This setup has been extended to incremental (Keswani et al., 2021) and sequential (Joshi et al., 2021) decision making, and to bandit feedback (Gao et al., 2021). Theoretical studies have analyzed the consistency (Mozannar & Sontag, 2020) and calibration (Verma & Nalisnick, 2022) of the HDM pipeline and the structure of optimal deferral policies (Okati et al., 2021).

**Issues with Separation of Responsibilities.** At a high level, an HDM strategy should satisfy the following desiderata:

**D1**. *Complementarity.* It should leverage the complementary capabilities of each agent to obtain better decisions *on average*, or equally good decisions at a lower cognitive cost, than each agent individually.

**D2**. *Synergy.* It should combine the contributions of each agent to obtain better *individual* decisions, or equally good decisions at a lower cognitive cost, than each agent individually.

**D3**. *Reliability.* It should produce decisions that are more reliable than those made by each agent individually.

Existing HDM approaches aim at enabling complementarity (**D1**). In fact, the main benefit of offloading decisions to an AI is that of lowering the human's cognitive effort. Moreover, depending on the relative performance of the predictor on inputs in $\mathcal{M}$ compared to the expert, they can also improve the quality of the team's decisions (on average across inputs, not necessarily for all inputs). Under suitable conditions, learning to defer can *provably* do so (Donahue et al., 2022). However, approaches complying with separation of responsibilities completely overlook synergy (**D2**). When the machine outputs a decision, the human is tempted to simply stick to it, thus *over-relying on the machine's decisions*, because of the previously mentioned anchoring (Rastogi et al., 2022) and automation (Cummings, 2012) biases. Conversely, whenever the machine opts for deferral, *the human is left resolving hard cases completely unassisted*. This also compromises reliability (**D3**), which is key in high-stakes applications, thus hindering the applicability of HDM.

## 3 Beyond Separation of Responsibilities with Learning to Guide

We propose *learning to guide* (LTG), a novel HDM framework that addresses the shortcomings of existing strategies by foregoing separation of responsibilities. In a nutshell, LTG aims to learn a *guidance generator* $\gamma(\mathbf{x})$, implemented as a machine learning model, that given an input $\mathbf{x}$, outputs guidance $g$ that is useful

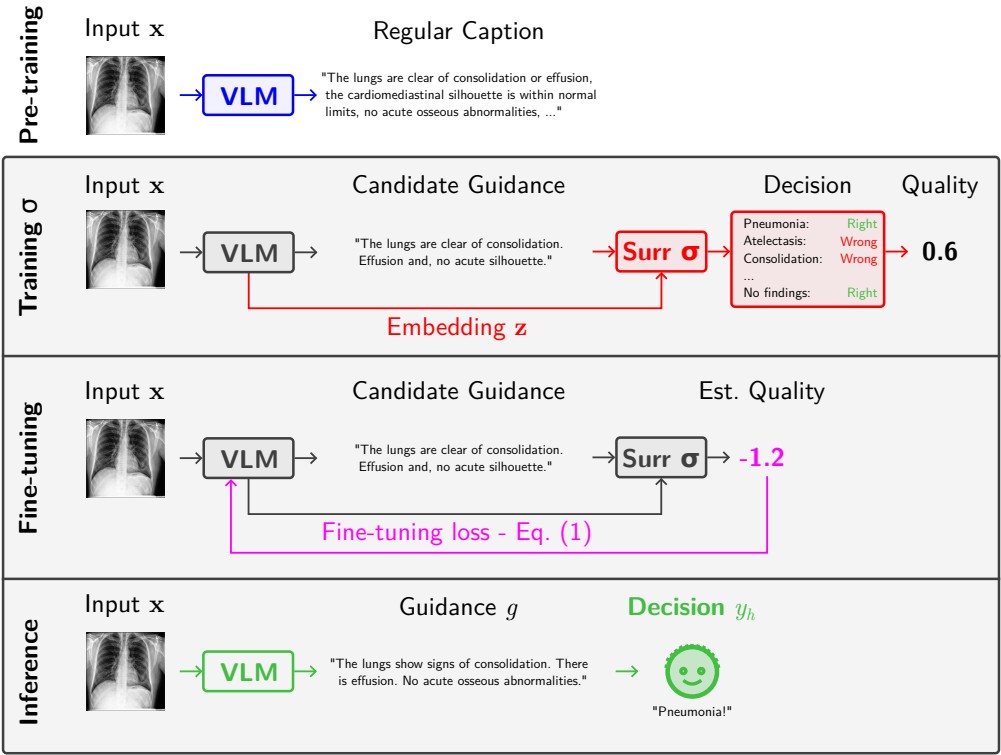

Figure 2: **The slog approach to learning to guide**. **Tier 1**: First we take a VLM (in **blue**) pre-trained to generate captions of visual inputs **x**. **Tier 2**: Next, we train the surrogate $\sigma_{\text{quality}}$ (in **red**) to estimate the quality of downstream decisions using a modicum of annotated guidance-quality pairs. The surrogate takes both images and text (embeddings) as input. **Tier 3**: Given a trained surrogate $\sigma_{\text{quality}}$, we fine-tune the VLM (in **magenta**) to output guidance $g$ achieving high (estimated) decision quality. **Tier 4**: The fine-tuned VLM (in **green**) can readily be used for generating useful textual guidance.

for *assisting human decision making* on that input. In medical diagnosis, for instance, given a chest X-ray scan **x**, the guidance $g$ might describe pathology visible in the image that are useful for identifying pathologies and prescribe treatment, as shown in Fig. 2 (top). Critically, and in stark contrast with existing HDM approaches, in LTG the machine does not replace the human: *the final decision is always taken by the human partner, in collaboration with the AI*. This means the decision maker is always in the loop and responsible for the final decision.

**Desiderata for Guidance**. In order to support human decision making, guidance should satisfy the following natural properties:

> **D4**. *Interpretability*. It should be *understandable* for the human expert at hand.

> **D5**. *Informativeness*. It should be *informative* for the decision at hand.

If these are satisfied, then guidance can be used by human experts to address a specific downstream decision making task. Note that, satisfying these desiderata encourages satisfaction of **D1**–**D3**. In fact, if guidance is interpretable (**D4**) and extracts decision-relevant elements from the input (**D5**), it should help the human in taking accurate decisions on individual instances (**D2**) and as a consequence improving the average quality of the decisions being made (**D1**). Additionally, interpretability helps the human in judging the quality of the guidance received, and thus evaluate the reliability of the overall decision (**D3**).

**Learning to Guide for VLMs**. In this paper we focus on *textual guidance* expressed in natural language as a mean to enable interpretability (**D4**). Motivated by their state-of-the-art performance in text generation

tasks (Wei et al., 2022) and by their promise in pathological report generation (Shamshad et al., 2023; Chen et al., 2020; 2021; Hou et al., 2021; Kayser et al., 2022; Yunxiang et al., 2023; Bazi et al., 2023; Drozdov et al., 2020; Yan & Pei, 2022), we propose to leverage *vision-language models* (VLM) to generate guidance.

Off-the-shelf VLMs are not conceived for generating guidance for *specific* decision making tasks, and thus violate informativeness (**D5**). Clearly, a perfectly accurate medical report is also an optimal guidance for follow-up decisions, but generating highly accurate reports requires massive amounts of supervision, and reports generated by specialized VLMs are far from perfect (Shamshad et al., 2023).

The question is then how to *convert* such models into high-quality guidance generators. Focusing on (medical) decision making from image data, we address this problem by introducing SLOG, a novel approach for *turning vision-language models into guidance generators using human feedback* designed to comply with **D1**–**D5**. The rationale behind SLOG is to encourage VLMs to focus on accurately reporting those aspects of the input image that are most relevant *for the follow-up decisions*, possibly overlooking less important details. Next, we briefly discuss how SLOG uses annotations and then proceed to outline the main algorithm.

### 3.1 Estimating Downstream Decision Quality

Optimizing guidance for synergy (**D2**) requires knowing the quality of downstream decisions taken by a human expert supplied with the guidance itself. SLOG assumes access to quality ratings $\mathbf{q} \in [0,1]^d$, where each $q_i$ encodes the quality of a downstream decision. For instance, if the expert has to determine the state of two conditions (e.g., "`pneumonia`" and "`fracture`"), then $d = 2$. Quality ratings for expert decisions can be obtained by comparing these against a gold standard (using, e.g., decision accuracy) or by consulting a second expert (using, e.g., a star rating system).

Clearly, there is a tension between the number of annotations necessary for fine-tuning a VLM and the cost of eliciting such annotations. SLOG addresses this issue by training a *surrogate model* $\sigma_{\text{quality}} : (\mathbf{x}, \mathbf{z}) \mapsto \widehat{\mathbf{q}}$ using a modicum of annotated quality ratings, and using it to estimate the quality of guidance $g$ generated by the VLM during fine-tuning. In practice, SLOG fits the surrogate on a training set $\mathcal{D}_{\text{surr}} = \{(\mathbf{x}_i, \mathbf{z}_i, q_i)\}$, where $\mathbf{x}_i$ is an input image, $\mathbf{z}_i$ is the embedding of the VLM's guidance $g_i$ for that input, and $q_i$ is the quality of that guidance, by minimizing an average cross-entropy loss of the form:

$$\frac{1}{|\mathcal{D}_{\text{surr}}|} \sum_{(\mathbf{x}, \mathbf{z}, \mathbf{q}) \in \mathcal{D}_{\text{surr}}} \frac{1}{d} \sum_{i=1}^{d} \mathsf{CE}(q_i, \sigma_{\text{quality}}(\mathbf{x}, \mathbf{z})_i) \tag{1}$$

### 3.2 The SLOG Loop

In essence, SLOG takes a pre-trained VLM caption generator $\gamma$ and fine-tunes it for a number of rounds $T$. Let $\mathcal{D}_{\text{train}}$ be a data set of image-caption pairs (for instance, a subset of the data that $\gamma$ was trained on) and $\mathcal{D}_{\text{tune}}$ a larger set of *unlabeled* images from the target decision making task. In each round $t = 1, \ldots, T$, SLOG samples a batch $\{\mathbf{x}_1, \ldots, \mathbf{x}_B\}$ from $\mathcal{D}_{\text{tune}}$ uniformly at random with replacement, and computes guidance $g_i^t = \gamma(\mathbf{x}_i)$ and embeddings $\mathbf{z}_i^t$ for each input. Then, it evaluates the quality of the generated guidance using the frozen surrogate $\sigma_{\text{quality}}$ and fine-tunes the VLM $\gamma$ by minimizing an augmented loss of the form:

$$\mathsf{CE}(\gamma, \mathcal{D}_{\text{train}}) - \frac{\lambda}{|\mathcal{D}_{\text{tune}}|} \sum_{(\mathbf{x}, \mathbf{z}) \in \mathcal{D}_{\text{tune}}} \frac{1}{d} \sum_{i=1}^{d} \sigma_{\text{quality}}(\mathbf{x}, \mathbf{z})_i \tag{2}$$

for a given number of epochs. Eq. (2) trades off text generation performance on the training set $\mathcal{D}_{\text{train}}$ – so as to discourage catastrophic forgetting – and estimated quality of downstream decisions on the fine-tuning set $\mathcal{D}_{\text{tune}}$. Here, $\lambda > 0$ is a hyper-parameter. Fine-tuning then amounts to applying gradient descent to batches comprising training and fine-tuning examples in equal proportions. Once done, the SLOG loop repeats. As long as the surrogate generalizes the quality rating annotations, the VLM gradually learns to output text that works well as guidance tailored for the target decision task.

### 3.3 Benefits and Limitations

In stark contrast with existing HDM strategies, SLOG ensures that the human receives guidance useful for decision making while keeping them in the loop. The cognitive load of LTG is entirely devoted to ensuring it can be safely employed in high-stakes applications, where there is little room for mistakes and humans *have* to be in control at all times (Zhang et al., 2020), as increasingly prescribed by legal frameworks (Government of Canada, 2019; European Commission, 2021). LTG and SLOG are designed explicitly for supporting HDM in these cases. SLOG is reminiscent of mainstream approaches to LLM alignment, such as reinforcement learning with human feedback (RLHF) (Ziegler et al., 2020; Ouyang et al., 2022), but differs from them in aims and technology. While RLHF strives to improve factuality and reduce harmfulness of generated content (Ouyang et al., 2022), ignoring the decisions these impinge on, SLOG specifically aims at improving quality of downstream human decisions for a specific decision making task. At the same time, SLOG foregoes reinforcement learning approaches (Schulman et al., 2017) in favor of a simpler and more direct end-to-end fine-tuning strategy.

One limitation of SLOG is that the performance of the guidance generator hinges on that of the surrogate, which in turn relies on the amount of quality ratings available for training. In Section 4 we present an ablation study showing that a limited amount of quality annotations are sufficient for SLOG to improve generated guidance. Another limitation of SLOG is that it currently assumes quality ratings are readily available, which is not always the case. One option is then to integrate SLOG with active learning strategies (Settles, 2012; Herde et al., 2021) to acquire informative quality ratings whenever needed. Doing so is however outside the scope of this paper and left to future work. Finally, the guidance output by VLMs may suffer from hallucination, that is, it may contain untrue statements. However, SLOG directly maximizes factuality of guidance on the fine-tuning set $\mathcal{D}_{\text{tune}}$, meaning that a simple way of reducing the chance of non-factual statements is to employ a larger fine-tuning set. This is relatively cheap to do, as no annotations are required. Moreover, large language models can be surprisingly well-calibrated (Kadavath et al., 2022), meaning that generated guidance can be filtered based on the VLM's own uncertainty estimates to prevent over-reliance (Eigner & Händler, 2024) and avoid low-quality decisions (Zhang et al., 2020).

## 4 Empirical Analysis

In this section, we investigate the following research questions:

**Q1** Does SLOG work in a controlled setting?

**Q2** Does SLOG improve generated guidance despite relying on a surrogate model?

**Q3** Does SLOG improve the quality of the decisions made using its guidance?

**Q4** Does SLOG help improve performance even when examined by a human physician?

In order to answer **Q1**, we investigate the effectiveness of SLOG in a controlled experimental setup using the ClevR dataset (Johnson et al., 2017). For the remaining questions, we employ a real world medical decision making dataset. We discuss the specifics of all experiments in the following. We will make all source code public upon acceptance of the manuscript.

### 4.1 Q1: The ClevR Task

**Data set**. We first evaluate SLOG on a variant of the CLEVR (Johnson et al., 2017) dataset. This consists of $60k$ images for training, $10k$ for validation and $15k$ for testing. Each image represents several three-dimensional objects with different shapes, colors, sizes, materials, and positions, and comes with a structured (non-textual) description of its contents. We translate these into natural language to obtain ground-truth textual descriptions, see Fig. 3 (left) for an example.

**Decision-making task**. We are interested in evaluating SLOG's ability of producing high-quality guidance for decision-makers. Since the CLEVR data set comes with no pre-specified decision task, we define one

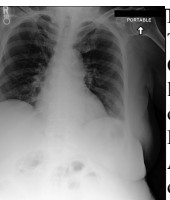

Figure 3: **Left**: Example CLEVR image and corresponding textual description. **Right**: Example of `Mimic-CXR-IV` radiograph and corresponding medical report, comprising *findings* and *impression* sections.

ourselves. Specifically, we construct the following six rules about the presence of certain objects in the image:

**Rule 1** Does the image contain one large green sphere **or** one small rubber cube?

**Rule 2** Does the image contain one large red sphere **or** one green object?

**Rule 3** Does the image contain one large rubber cube **and** one sphere?

**Rule 4** Does the image contain one rubber cylinder **and** two small objects?

**Rule 5** Does the image contain one small red metal cube **or** two rubber cylinders?

**Rule 6** Does the image contain one sphere **and** two small metal objects?

Each rule allows us to define positive images (i.e., those that satisfy the rule) and negative images (i.e., those that do not), yielding six labels per image. We construct these rules based on the frequency of object features in the images, selecting conditions with a balanced positive/negative ratio. Given an image, the decision task amounts to predicting what rules will fire.

In a nutshell, in this experiment we first train a VLM on the ground-truth (decision-agnostic) captions, then construct a surrogate model that simulates the human decision-maker, and then fine-tune the VLM with SLOG (Equation (2)) to produce guidance. Next, we detail each part of the pipeline.

**Vision-language Model**. We apply SLOG to a vision encoder-decoder model with a simple transformer (Vaswani et al., 2017) model. Given an image $\mathbf{x}$, the VLM outputs a textual caption describing what objects it contains. We trained the model with an nVidia A100 40 GB GPU with a batch size of 256, restricting the maximum length to 55. With a patience of 25, we trained the model for 30 epochs and retained the model achieving the best BLEU$_4$ score. The BLEU$_k$ score is useful to evaluate the quality of machine-generated text with respect to a reference text, usually generated by a human. Specifically, BLEU$_k$ considers the overlap of each $k$-gram between the machine-generated and the reference text (Papineni et al., 2002).

**Simulating the human expert**. To retain full control over the experimental setting, we *simulate* the human expert using a machine learning classifier, denoted HUMANPROXY, which takes as input an image $\mathbf{x}$ and the corresponding guidance $g$ and uses them to infer the six binary labels. The surrogate HUMANPROXY is a multi-modal model with a RESNET101 model as visual extractor (He et al., 2016) and a transformer module as textual encoder. The sole purpose of this surrogate is to produce (possibly imperfect) decisions for all six rules for the entire data set, which are not provided by the original CLEVR dataset but are needed for training the SLOG surrogate model $\sigma_{\text{quality}}$.

Specifically, we obtain the prediction using a $k$-fold cross validation procedure: in each fold, we fit HUMAN-PROXY using the ground-truth labels (rule activations) from the training split, and produce predictions for all examples in the test split. Then we aggregate all test prediction to annotate the whole datasets. This step ensures there is no data leakage between train and test splits. In our experiental set-up, we considered $k = 5$.

**Quality surrogate model**. The surrogate $\sigma_{\text{quality}}$ takes the same inputs as HUMANPROXY, but estimates the quality of the predictions made by HUMANPROXY. To this end, we compare said predictions with the

Table 1: SLOG **boosts estimated quality of generated guidance without compromising text quality in** CLEVR. The results show that SLOG substantially improves estimated guidance quality as measured by the surrogate model ($\sigma_{\text{quality}}$) without affecting text quality as measured by BLEU scores over ground-truth caption data.

| MODEL | BLEU$_1$ | BLEU$_2$ | BLEU$_3$ | BLEU$_4$ | BLEURT | $\sigma_{\text{quality}}$ |
|---|---|---|---|---|---|---|
| Baseline | 0.95 | 0.92 | 0.89 | 0.85 | 0.73 | 1.96 |
| Fine-tuned | 0.94 | 0.92 | 0.89 | 0.85 | 0.72 | 1.95 |
| SLOG | **0.98** | **0.96** | **0.93** | **0.88** | **0.76** | **2.28** |

Table 2: SLOG **boosts the quality on the downstream decision in** CLEVR, as shown by the precision, recall and $F_1$ performance of decisions entailed by SLOG's guidance compared to that of decisions entailed by the baseline VLM and a VLM fine tuned without the SLOG loss. Best $F_1$ results in **bold**.

| Rule | Baseline | | | Baseline (Fine-tuned) | | | SLOG | | |
|---|---|---|---|---|---|---|---|---|---|
| | Pr | Rc | $F_1$ | Pr | Rc | $F_1$ | Pr | Rc | $F_1$ |
| Rule 1 | 92.33 | 95.91 | 94.09 | 93.55 | 94.03 | 93.79 | 94.52 | 97.71 | **96.09** |
| Rule 2 | 99.98 | 97.47 | 98.71 | 100.00 | 95.84 | 97.88 | 99.92 | 99.05 | **99.49** |
| Rule 3 | 99.75 | 84.93 | 91.75 | 99.75 | 84.89 | 91.72 | 99.77 | 90.09 | **94.68** |
| Rule 4 | 94.44 | 96.67 | 95.54 | 96.74 | 94.63 | 95.67 | 96.97 | 98.33 | **97.65** |
| Rule 5 | 95.47 | 93.38 | 94.41 | 98.05 | 92.25 | 95.06 | 97.91 | 96.25 | **97.07** |
| Rule 6 | 86.59 | 98.26 | 92.05 | 93.88 | 94.70 | 94.28 | 93.36 | 99.45 | **96.31** |
| MACRO AVG | 94.76 | 94.44 | 94.42 | 97.00 | 92.72 | 94.73 | 97.07 | 96.82 | **96.88** |
| MICRO AVG | 94.42 | 95.14 | 94.78 | 96.84 | 93.32 | 95.05 | 96.94 | 97.31 | **97.12** |

ground-truth rule activations and record whether they match or not. Then we train $\sigma_{\text{quality}}$ to predict the *correctness* of predictions given only the image $\mathbf{x}$ and guidance $g$. We employed 70%-20%-10% splits for training, validation and test, respectively. We trained $\sigma_{\text{quality}}$ with a cross entropy loss to predict the accuracy of the predictions, and selected the model that attained the best micro validation $F_1$. We set patience to 20 epochs, for a total of 75 training epochs.

**Applying** SLOG. Given the trained surrogate $\sigma_{\text{quality}}$, we ran the SLOG finetuning for 10 epochs. In this phase, we froze both $\sigma_{\text{quality}}$ and the visual encoder of the VLM. We evaluated several values of the hyperparameter $\lambda$ (cf. Eq. (2)) and chose $\lambda = 1$ as it yielded the highest $F_1$ score on a validation split.

**Competitors and metrics**. We assess the performance of SLOG both quantitatively and qualitatively, and compare it against two competitors. These include the original pretrained VLM (denoted "baseline") as well as the same VLM fine-tuned for the same number of epochs as the SLOG variant, but with $\lambda = 0$, so as to disable the SLOG loss term (denoted "fine-tuned").

We evaluate both the quality of down-stream decisions taken using the generated guidance and the textual guidance itself. For the former, we report the test set precision (Pr), recall (Rc), and $F_1$ score of the decisions, both per-label and (micro- and macro-) averaged over all six labels. We obtain the down-stream decision by applying the rules to the generated guidance, using a simple NLP pipeline that scans textual guidance for presence of individual objects and their properties (e.g., "a large red sphere") and checks which rules fire based on the patterns it matched. We compare the resulting decisions against the ground-truth labels. For the latter, we report the $BLEU_k$ score for $k = 1, \ldots, 4$ with respect to the ground-truth captions, the average guidance length, and the average estimated quality output by $\sigma_{\text{quality}}$ as a sanity check.

**Q1:** SLOG **improves the quality of downstream decisions in our controlled setting**. Our results are summarized in Tables 1 and 2. Table 1 shows that SLOG's guidance yields a distinct improvement in terms of $\sigma_{\text{quality}}$ score (second to last column), as expected. This highlights how the SLOG fine-tuning procedure succeeds in optimizing the SLOG loss (Eq. (2)). The remaining results suggest that doing so yields better

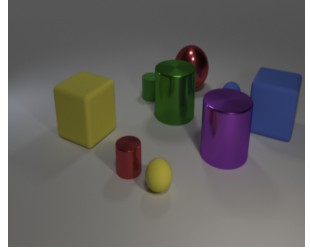

**GT.**1 small blue rubber sphere 1 large purple metal cylinder 1 small red metal cylinder 1 large yellow rubber cube 1 small green rubber cylinder 1 large green metal cylinder 1 large blue rubber cube 1 large red metal sphere 1 small yellow rubber sphere .
**Base.**1 small blue rubber sphere 1 small red metal cylinder 1 large purple metal cylinder 1 small yellow rubber sphere 1 large blue rubber cube 1 large yellow rubber cube 1 small green rubber cylinder 1 large green metal cylinder .
**FT.**1 small red metal cylinder 1 small blue rubber sphere 1 large red metal sphere 1 large yellow rubber cube 1 small yellow rubber sphere 1 large purple metal cylinder 1 large green metal cylinder 1 small green rubber cylinder .
**SLOG.**1 small red metal cylinder 1 small blue rubber sphere **1 large red metal sphere** 1 large green metal cylinder 1 large yellow rubber cube 1 small yellow rubber sphere 1 large purple metal cylinder 1 small green rubber cylinder **1 large blue rubber cube**.

Figure 4: **Example guidance generated by different competitors on** CLEVR**.** While SLOG neither misses nor hallucinates any description, each of the other two competitors, despite not hallucinating, misses one description. The missed descriptions are presented in green bold fonts in the above text.

Table 3: Comparison between the original and our filtered splits for the `Mimic-CXR-IV` dataset.

| COMPONENTS | ORIGINAL SPLIT | | | OUR SPLIT | | |
|---|---|---|---|---|---|---|
| | TRAIN | VAL | TEST | TRAIN | VAL | TEST |
| REPORTS | 222,758 | 1,808 | 3,269 | 125,417 | 991 | 1,624 |
| IMAGES | 368,960 | 2,991 | 5,159 | 232,855 | 1,837 | 2,872 |

guidance. In fact, SLOG also out-performs both competitors in terms of textual quality: it achieves between +3 and +4 % in terms of BLEU scores and the BLEURT[2] (Sellam et al., 2020) metric (first five columns) compared to both the pretrained model and its fine-tuned variant.

Simultaneously, Table 2 indicates that the guidance produced with SLOG facilitates inferring correct labels compared to the captions produced by the baseline models. This holds for each rule/label individually (first six rows), and on average (last two rows). This provides initial evidence that SLOG also improves the quality of down-stream decisions in a controlled setting.

To further illustrate the benefits of SLOG, we report in Figure 4 an example of the guidance it produces, compared to the captions output by the competitors. The example shows how SLOG's guidance describes objects that are relevant for the decision correctly that the competitors neglect.

### 4.2 Q2–Q4: The `Mimic-CXR-IV` Task

**Data set**. Next, we evaluate SLOG on the `Mimic-CXR-IV` data set (Johnson et al., 2019), one of the largest publicly available medical decision data sets, consisting of $227,835$ radiology reports and $377,110$ chest X-ray scans. We focus on the *findings* and the *impression* sections of the reports. As shown in Fig. 3 (right) the findings are text-based descriptions of what can be observed in the scan, and constitute the basis on top of which the expert forms their impression, i.e., their initial opinion about the potential pathologies of a patient. We discarded examples where either findings or impressions were not available, resulting in the *training*, *validation* and *test* splits presented in Table 3.

**Vision-language Models**. We apply SLOG to three vision-language models: `R2Gen` (Chen et al., 2020), `R2GenCMN` (Chen et al., 2022) and, `R2GenMamba` (Sun et al., 2025). `R2Gen` is a memory-driven transformer (Vaswani et al., 2017) specifically designed for pathological report generation from chest X-ray images, while `R2GenCMN` uses a cross modal network (CMN) in order to achieve better mapping between diverse modalities. Finally, `R2GenMamba` leverages a recent MAMBA state-space model in its encoder architecture. The `R2Gen` architecture builds on the observation that similar radiographs may correspond to reports sharing similar patterns. To exploit this, it employs a pre-trained CNN model to extract patch features and encodes these into hidden states with an encoder. A decoder then maps the hidden states into words at each time point with the help of a relational memory and memory-driven conditional layer normalization. The relational

---

[2]While the BLEU scores measure the $k$-gram based overlap between the predicted and generated texts, the BLEURT is a BERT (Devlin et al., 2019) based regression model trained on human-ratings data.

memory component allows the transformer to store and repurpose shared patterns and thus generate more coherent reports (Chen et al., 2020). Chen et al. (2022) argues that the existing literature offers only limited scope for proper alignment across modalities. Addressing this issue, the authors developed `R2GenCMN`, a cross modal network where the encoded features of an image is fed to the CMN module to obtain the memory representations. A similar operation is done for the text embeddings. Thus, the shared information of the text and visual features can be stored in the memory. In particular, the CMN module employs a matrix where each row of the matrix is allotted for cross-modal memory information for image and texts.

**Decision-making task**. Our real-world experiment focuses on a critical step of the medical decision process: making the right diagnosis.The task is to diagnose 14 different pathologies (see Table 6 for the full list) from X-ray images. In our experiments, we pre-train the `R2Gen` and `R2GenCMN` VLMs to predict *findings* from images, and then fine-tune them with SLOG to improve their generated guidance. Given that the *impression* is the opinion that the expert forms about potential pathologies visible in the image, *we fine-tune our VLMs to produce textual guidance that – once interpreted by a human expert – leads to the same diagnosis entailed by the impression.*

**Simulating the human expert**. As in the case of CLEVR task, we simulate human decisions using a machine learning model denoted HUMANPROXY, for reproducibility. Specifically, HUMANPROXY takes a scan $\mathbf{x}$ and a corresponding VLM-generated report and diagnoses the 14 candidate pathologies using three classes: definitely present (*positive*), definitely absent (*negative*), and unclear (*ambiguous*). Following (Lovelace & Mortazavi, 2020a), we implement HUMANPROXY as a classification model that takes both reports and images as inputs and train it on ground-truth labels obtained by applying the `CheXpert` (Irvin et al., 2019) automated annotation tool to the ground-truth *impressions*. Please note that while the $\sigma_{\text{quality}}$ surrogate is an integral part of SLOG, HUMANPROXY is an experimental detail necessary for evaluation.

In order to emulate a setting with sparse human supervision, we assume *ground-truth labels are available for 10% of the training data only.* This ground-truth dataset $\mathcal{D}_{\text{surr}}$ is used for training both the model simulating human decisions HUMANPROXY and the quality surrogate model $\sigma_{\text{quality}}$ estimating the quality of these decisions. We rely on a stratified sampling procedure to select $\mathcal{D}_{\text{surr}}$ so as to maintain a reasonable coverage of the different classes.

Overall, we proceed as follows. First, HUMANPROXY is trained on $\mathcal{D}_{\text{surr}}$ to output a diagnosis given ground-truth findings (as these are the only ones for which we know the corresponding human diagnosis). Once trained, we use HUMANPROXY to produce quality ratings by computing the correctness of its predictions over $\mathcal{D}_{\text{surr}}$, which will later on be used for training the surrogate $\sigma_{\text{quality}}$.

To avoid biasing the quality rating supervision by computing it on training instances, we run *k*-fold cross validation on $\mathcal{D}_{\text{surr}}$ and collect quality ratings from the *k* validation folds. For each validation fold, we compute decisions using both VLM-generated guidance and ground-truth text, so as to provide examples of both predicted and ground-truth guidance to train the quality surrogate model.

**Quality surrogate model**. As explained in Section 3.1, the surrogate $\sigma_{\text{quality}}$ should estimate the quality of human decisions when fed with the VLM guidance. In this experiment, human decisions are proxied with the 14 labels output by HUMANPROXY. The surrogate is thus trained to predict the *correctness* of each of the 14 predictions made by HUMANPROXY. Just like HUMANPROXY, the surrogate $\sigma_{\text{quality}}$ used by SLOG is also implemented as mutli-modal architecture and trained to minimize the average cross entropy loss on the ground-truth dataset described in the previous paragraph. Albeit having different purposes, HUMANPROXY and $\sigma_{\text{quality}}$ share the same model architecture. The multimodal functionality of both the models are established with a visual-encoder module and a text encoder module, where the former is a ResNet 101 based model that extracts the features of the radiology images and the latter is a transformer based module that extracts nuanced features of the reports. To this end, we concatenate the features obtained from the text extractor and image extractor before applying a fully connected layer onto the concatenation.

**Overall pipeline**. First, an `R2Gen` VLM is pre-trained on the training split $\mathcal{D}_{\text{train}}$ to generate findings. The VLM is then applied to the decision ground-truth dataset $\mathcal{D}_{\text{surr}}$ (10% of the training split). The generated guidance is fed to HUMANPROXY to obtain (simulated) human decisions and corresponding quality annotations. This information is then used to fit the quality surrogate model $\sigma_{\text{quality}}$. Finally, the VLM is

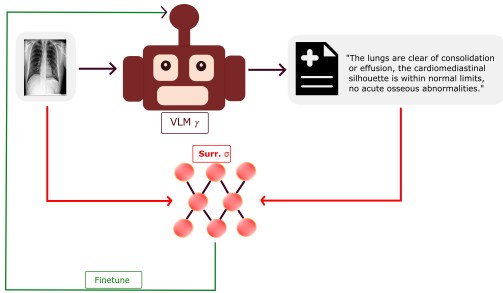

Figure 5: Finetuning policy of SLOG.

fine-tuned *on the entire training split* according to Eq. (2) for 10 epochs and evaluated on the test data. The source code is provided in the supplementary material. Fig. 5 depicts a summary of the overall pipeline.

**Training the VLM**. We trained our baseline VLM with an Nvidia A100 80GB GPU and with batch size 256. We restricted the maximum sequence length to 70 in order to avoid computational over head in later stages of our experiment. We fine-tuned the baseline models (`R2Gen` and `R2GenCMN`) on the same GPU with a batch size of 64. The values of all hyper-parameters were taken verbatim from (Chen et al., 2020) and (Chen et al., 2021), respectively. While training the baseline model, we used a patience of 20 and stored the best model with the highest BLEU-4 score.

**Training the surrogate**. In order to emulate a setting with sparse human supervision, we assume *ground-truth labels are available for 10% of the training data only*. This ground-truth dataset $\mathcal{D}_{\text{surr}}$ is used for training both the model simulating human decisions HUMANPROXY *and* the quality surrogate $\sigma_{\text{quality}}$ estimating the quality of these decisions. HUMANPROXY, which acts as a proxy for the human annotator, is a multimodal classification model which takes the radiology image and corresponding report as inputs and predicts the 14 target symptoms (cf. Table 6). To this end, in addition to calculating the loss and classification metrics, we conducted a sample-wise comparison between the ground truth labels and the prediction with the purpose of generating training data for $\sigma_{\text{quality}}$. This comparison yielded an $m \times n$ matrix $Y_q$ where $m = 14$ and $n$ is the number of training examples. Let us consider $G$ is the ground truth matrix of labels used for HUMANPROXY and $P$ is the matrix of labels predicted by the model on the validation data. We define,

$$Y_q = y_{ij}, \quad \text{where } y_{ij} = \begin{cases} 1 & \text{if } G_{ij} = P_{ij} \\ 0 & \text{Otherwise} \end{cases} \tag{3}$$

In order to train HUMANPROXY, we use k-fold cross validation with $k = 5$. We assessed the performance of our model on the validation set by scrutinizing the micro $F_1$ score pertaining to the positive mentions.

The $\sigma_{\text{quality}}$ takes same input as the HUMANPROXY, but instead of three labels, it outputs either 1 or 0 for the 14 classes (see Equation 3). In Table 4, we report the results obtained from the test split that was used to evaluate the performance of $\sigma_{\text{quality}}$. Results clearly indicate that $\sigma_{\text{quality}}$ is capable of reliably predicting the correctness of HUMANPROXY when provided image and guidance.

**Applying** SLOG. We apply SLOG finetuning for 10 epochs. In this phase, we freeze both the $\sigma_{\text{quality}}$ and the visual encoder layer of the baseline `R2Gen` model. We try with varying values of $\lambda$ and the best model was chosen based on the validation $F_1$ score. Eventually, for `R2Gen`, we choose 10 as the value of hyperparameter $\lambda$ as $\lambda = 10$ yielded the best $F_1$ during finetuning. Along with finetuning using $\lambda = 10$, we also experiment with $\lambda = 0$ to finetune without the $\sigma_{\text{quality}}$. In both cases, we finetune the baseline model for equal number of epochs. We follow the same pipeline for `R2GenCMN` and chose 0.01 as the value for hyperparameter $\lambda$.

**Q2:** SLOG **improves informativeness on the test set without compromising BLEU score.** A potential issue with using a surrogate model as a proxy of decision quality is that the fine-tuned VLM might end up overfitting the surrogate and produce guidance that, while seemingly informative, is unrelated to the actual input scan. SLOG prevents this by complementing estimated guidance quality as computed by the surrogate model with guidance appropriateness for the input image as measured by cross entropy over a training set of findings. Table 5 confirms the effectiveness of this strategy. SLOG substantially improves

Table 4: **Outcomes from the test split used to evaluate** $\sigma_{\text{quality}}$. Results of `R2Gen` and `R2GenCMN` showing per-class, macro, and micro averaged precision ($Pr$), recall ($Rc$), and $F_1$ scores.

| PATHOLOGY | R2Gen | | | R2GenCMN | | |
|---|---|---|---|---|---|---|
| | $Pr$ | $Rc$ | $F_1$ | $Pr$ | $Rc$ | $F_1$ |
| No Findings | 54.88 | 87.53 | 67.46 | 86.22 | 79.69 | 82.82 |
| Cardiomediastinum | 84.68 | 93.47 | 88.85 | 92.65 | 93.24 | 92.94 |
| Cardiomegali | 90.24 | 94.11 | 92.13 | 94.25 | 94.65 | 94.45 |
| Lung Lesion | 78.28 | 93.07 | 85.03 | 91.93 | 91.89 | 91.91 |
| Lung Opacity | 92.07 | 94.21 | 93.13 | 93.61 | 95.48 | 94.54 |
| Edema | 92.52 | 93.87 | 93.19 | 94.99 | 96.82 | 95.90 |
| Consolidation | 91.99 | 92.71 | 92.35 | 93.12 | 94.65 | 93.88 |
| Pneumonia | 42.08 | 84.86 | 56.26 | 74.99 | 75.51 | 75.25 |
| Atelectasis | 82.87 | 94.67 | 88.38 | 93.58 | 92.36 | 92.97 |
| Pneumothorax | 60.89 | 91.55 | 73.13 | 87.55 | 85.46 | 86.49 |
| Pleural Effusion | 96.55 | 97.12 | 96.84 | 96.73 | 98.38 | 97.55 |
| Pleural Other | 62.86 | 87.97 | 73.33 | 84.00 | 81.90 | 82.94 |
| Fracture | 93.94 | 93.86 | 93.90 | 93.62 | 94.36 | 93.99 |
| Support Devices | 78.03 | 91.75 | 84.34 | 91.46 | 88.19 | 89.80 |
| MACRO | 92.96 | 80.84 | 86.19 | 90.62 | 90.18 | 90.39 |
| MICRO | 92.20 | 78.71 | 84.17 | 91.18 | 90.84 | 91.01 |

Table 5: SLOG **boosts estimated quality of generated guidance without compromising text quality**. The results show that SLOG substantially improves estimated guidance quality as measured by the surrogate model ($\sigma_{\text{quality}}$) without affecting text quality as measured by BLEU scores over ground-truth caption data.

| MODEL | SETTING | BLEU$_1$ | BLEU$_2$ | BLEU$_3$ | BLEU$_4$ | BLEURT | $\sigma_{\text{quality}}$ |
|---|---|---|---|---|---|---|---|
| R2Gen | Pretrained | **0.36** | **0.22** | **0.15** | **0.11** | **-0.38** | 0.39 |
| | Fine-tuned | 0.33 | 0.21 | 0.14 | 0.10 | -0.40 | 0.39 |
| | SLOG | 0.35 | **0.22** | **0.15** | **0.11** | **-0.38** | **0.44** |
| R2GenCMN | Pretrained | **0.38** | **0.23** | **0.16** | **0.11** | -0.36 | 1.84 |
| | Fine-tuned | 0.37 | 0.22 | 0.15 | **0.11** | -0.35 | 1.85 |
| | SLOG | **0.38** | **0.23** | **0.16** | **0.11** | **-0.34** | **2.02** |
| R2GenMamba | Pretrained | 0.34 | 0.21 | 0.15 | **0.11** | -0.46 | 2.10 |
| | Fine-tuned | 0.35 | 0.22 | 0.15 | **0.11** | -0.38 | 2.11 |
| | SLOG | **0.35** | **0.22** | **0.15** | **0.11** | **-0.37** | **2.12** |

estimated guidance quality (second term in Eq. (2)) without compromising text quality, as measured both in terms of BLEU and BLEURT scores over test examples. For the sake of fairness, we compared SLOG (with $\lambda = 10$) with both the pre-trained `R2Gen` model (before the fine-tuning stage), and the `R2Gen` model fine-tuned for the same number of epochs as SLOG, but with caption-level supervision only (i.e., setting $\lambda = 0$ in Eq. (2)), as well as with two `R2GenCMN` and `R2GenMamba` models, respectively fine-tuned in the same way. Fig. 6 shows a qualitative example of the improvement in guidance of SLOG with respect to the competitors. First, SLOG's guidance retains all pieces of text that any of the other approach shares with the ground-truth text (green text). On top of this, SLOG retrieves additional chunks of text that are shared with ground truth findings (blue text) and impression (magenta text), even if the latter are never explicitly included as training

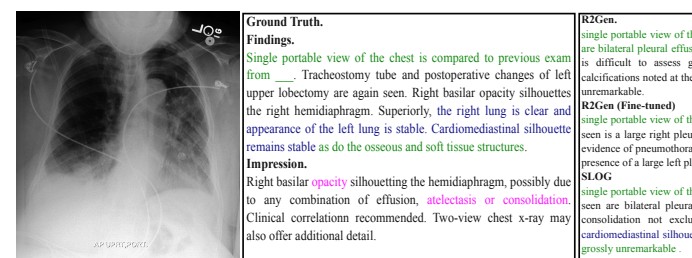

Figure 6: A qualitative example of the improvement of the guidance from SLOG with respect to the competitors. Green text indicates sentences that are (approximately) shared between the ground truth, SLOG and at least one of the competitors. Blue text indicates sentences shared between the ground truth findings (resp. impression) and SLOG, but missed by the competitors. No ground-truth sentences are shared between ground-truth text and competitors but missed by SLOG in this example.

supervision, confirming the effectiveness of the quality surrogate in encouraging the generation of relevant guidance for the diagnosis.[3]

**Q3: SLOG improves quality of decisions.** Tables 6 to 8 show the results in terms of decision quality, as measured by the $F_1$ score of the positive label for all 14 classes (multi-label prediction). Results clearly indicate the effectiveness of the SLOG guidance in improving decision quality, despite the modest amount of supervision it received. On R2Gen, SLOG outperforms the competitors in 8 out of 14 classes. All methods fail to identify any positive occurrence for three particularly under-represented classes (Pneumonia, Pleural Other, Fracture), while SLOG slightly under-performs with respect to pre-trained R2Gen in 2 classes only. It is worth noticing that SLOG is especially effective in improving recall without affecting precision on average, as shown by the two bottom lines reporting results averaged over classes (macro) and instances (micro) respectively. On R2GenCMN, SLOG is the best performing method on 6 classes, while the fine-tuned baseline wins on 5 classes, leaving the rest 3 as a tie among the three competitors. Additionally SLOG helps in recovering few remote cases that are missed with both the pretrained and the finetuned models, for example, 'Pneumonia' for R2Gen and R2GenCMN. The SLOG model developed on R2GenMamba, yields highest $F_1$ in 9 classes, while for the rest of the 5 classes, all the competitors fail to yield correct predictions. SLOG eventually yields highest $F1$ both in terms of 'micro' and 'macro' measures compared to its competitors. We also observe similar performance improve of SLOG comapared to its competitors for all the three models when evaluated with ChexBert (Smit et al., 2020). The results are added to the Tables 13 to 15 in Appendix A.

**Q4: SLOG has the potential to help doctors make better decisions.** The guidance quality of SLOG was evaluated against a fine-tuned R2GenCMN via expert review by a pulmonologist with three years of clinical experience. In this setup, the clinician assessed 55 guidances from SLOG and fine-tuned R2GenCMN (total 50). We decided to focus on fine-tuned R2GenCMN, the runner-up according to the previous experiment, to avoid overloading the clinician with too many assessments. It is important to examine whether SLOG helps the physician identify the presence of a symptom. Therefore, to ensure representative coverage while maintaining balanced class counts, we implemented a sampling technique that guarantes the presence of a symptom with at least one positive mention in the sample set. Thus, the samples cover all classes with at least one positive mention. The actual algorithm has been provided in the Appendix A. To this end, the entire task of annotation was randomized and blinded for the physician. In the sense, the physician, received an unordered representation of the reports and received no clue about the model that is responsible for a particular guidance. In order to ensure a fair competition between SLOG and its competitor, the physician was asked to annotate the same studies for both the models. Table 9 shows the results in terms of decision quality, measured by the $F_1$ score of the positive label for the 14 classes (prediction with multiple labels). The results confirm the advantage of SLOG in improving the overall performance of the decision-making process, both in terms of micro and macro $F_1$. In order to validate the results from 55 samples, the mean $F_1$ scores for each of the labels has also been reported. While SLOG falls short on 4 classes including 'No finding',

---

[3]Notice that while the SLOG guidance text is longer than the one of the competitors in this example, with various chunks of text which are not obviously connected to ground-truth ones, its overall quality is still much higher.

Table 6: Performance of `R2Gen` finetuned with SLOG. Results show per-class, macro and micro averaged precision, recall and $F_1$. Best $F_1$ results are boldfaced.

| PATHOLOGY | R2Gen | | | R2Gen (fine-tuned) | | | SLOG | | |
|---|---|---|---|---|---|---|---|---|---|
| | $Pr$ | $Rc$ | $F_1$ | $Pr$ | $Rc$ | $F_1$ | $Pr$ | $Rc$ | $F_1$ |
| No Finding | 37.94 | 65.84 | 48.14 | 38.82 | 65.61 | **48.78** | 39.74 | 62.22 | 48.50 |
| Cardiomediastinum | 0.0 | 0.0 | 0.0 | 0.0 | 0.0 | 0.0 | 0.0 | 0.0 | 0.0 |
| Cardiomegaly | 18.49 | 37.77 | 24.83 | 17.21 | 39.36 | 23.95 | 19.49 | 36.70 | **25.46** |
| Lung Lesion | 0.0 | 0.0 | 0.0 | 0.0 | 0.0 | 0.0 | 0.0 | 0.0 | 0.0 |
| Lung Opacity | 31.58 | 13.20 | 18.62 | 36.31 | 14.91 | 21.14 | 33.68 | 15.65 | **21.37** |
| Edema | 51.45 | 22.83 | 31.63 | 54.47 | 21.54 | 30.88 | 53.44 | 22.51 | **31.67** |
| Consolidation | 0.0 | 0.0 | 0.0 | 2.44 | 1.41 | 1.79 | 6.25 | 2.82 | **3.88** |
| Pneumonia | 0.0 | 0.0 | 0.0 | 0.0 | 0.0 | 0.0 | 50.00 | 0.65 | **1.29** |
| Atelectasis | 21.45 | 27.43 | 24.08 | 21.14 | 27.88 | 24.05 | 21.57 | 32.74 | **26.01** |
| Pneumothorax | 0.0 | 0.0 | 0.0 | 26.67 | 13.33 | **17.78** | 23.53 | 13.33 | 17.02 |
| Pleural Effusion | 66.82 | 38.74 | 49.04 | 68.54 | 40.11 | 50.61 | 62.91 | 52.20 | **57.06** |
| Pleural Other | 0.0 | 0.0 | 0.0 | 0.0 | 0.0 | 0.0 | 0.0 | 0.0 | 0.0 |
| Fracture | 0.0 | 0.0 | 0.0 | 0.0 | 0.0 | 0.0 | 0.0 | 0.0 | 0.0 |
| Support Devices | 28.24 | 54.55 | 37.21 | 28.40 | 53.41 | 37.08 | 28.41 | 55.68 | **37.62** |
| MACRO | 18.28 | 18.60 | 16.68 | 21.00 | 19.83 | 18.29 | 24.22 | 21.04 | **19.28** |
| MICRO | 33.03 | 31.40 | 32.19 | 33.17 | 31.96 | 32.55 | 33.98 | 33.84 | **33.91** |

Table 7: Performance of `R2GenCMN` finetuned with SLOG. Results show per-class, macro and micro averaged precision, recall and $F_1$. Best $F_1$ results are boldfaced.

| PATHOLOGY | R2GenCMN | | | R2GenCMN (fine-tuned) | | | SLOG | | |
|---|---|---|---|---|---|---|---|---|---|
| | $Pr$ | $Rc$ | $F_1$ | $Pr$ | $Rc$ | $F_1$ | $Pr$ | $Rc$ | $F_1$ |
| No Finding | 38.19 | 48.64 | 42.79 | 40.24 | 54.07 | **46.14** | 38.72 | 50.45 | 43.81 |
| Cardiomediastinum | 0.0 | 0.0 | 0.0 | 0.0 | 0.0 | 0.0 | 0.0 | 0.0 | 0.0 |
| Cardiomegaly | 16.33 | 55.85 | 25.27 | 18.30 | 59.57 | **28.00** | 17.73 | 53.19 | 26.60 |
| Lung Lesion | 0.0 | 0.0 | 0.0 | 0.0 | 0.0 | 0.0 | 33.33 | 1.67 | **3.17** |
| Lung Opacity | 29.55 | 27.38 | 28.43 | 30.74 | 19.32 | 23.72 | 33.81 | 29.10 | **31.27** |
| Edema | 48.33 | 9.32 | 15.63 | 48.21 | 17.36 | 25.53 | 54.03 | 21.54 | **30.80** |
| Consolidation | 15.71 | 15.49 | 15.60 | 12.50 | 16.90 | 14.37 | 14.61 | 18.31 | **16.25** |
| Pneumonia | 0.0 | 0.0 | 0.0 | 15.38 | 2.61 | **4.47** | 14.29 | 1.96 | 3.45 |
| Atelectasis | 17.60 | 30.53 | 22.33 | 22.12 | 31.42 | **25.96** | 18.52 | 30.97 | 23.18 |
| Pneumothorax | 16.67 | 10.00 | 12.50 | 30.00 | 10.00 | 15.00 | 28.57 | 13.33 | **18.18** |
| Pleural Effusion | 71.43 | 34.34 | 46.38 | 63.90 | 35.99 | 46.05 | 64.44 | 39.84 | **49.24** |
| Pleural Other | 0.0 | 0.0 | 0.0 | 0.0 | 0.0 | 0.0 | 0.0 | 0.0 | 0.0 |
| Fracture | 0.0 | 0.0 | 0.0 | 0.0 | 0.0 | 0.0 | 0.0 | 0.0 | 0.0 |
| Support Devices | 25.05 | 68.18 | 36.64 | 27.27 | 64.77 | **38.38** | 26.19 | 65.91 | 37.48 |
| MACRO | 19.92 | 21.41 | 17.54 | 22.05 | 22.29 | 19.12 | 24.59 | 23.31 | **20.25** |
| MICRO | 27.77 | 31.52 | 29.53 | 30.08 | 32.72 | 31.34 | 29.98 | 34.40 | **32.04** |

'Atelectasis', 'Pneumothorax' and 'Pleural Effusion', it outperforms the baseline on 7 classes. Additionally, in Table 10 we examined the pattern of error of the two models based on their confusion matrix transitions (Baseline → SLOG). We focused on the pathologies responsible for largest inflation in error, i.e. $\Delta FP + \Delta FN$. In particular 'Lung lesion' ($+10$ FP), 'Atelectasis' ($+6$ TP, $-1$ TP) and Pleural Effusion ($+4$ FP, $-1$ TP). It is worth noticing that the dominant source of error is due to over predicting, rather than missing true pathologies. Furthermore, rare instances of positive $\Delta$ in *false negatives* bolsters the fact that the lower performance of SLOG majorly stems from over detection, rather than missing actual cases  a phenomenon often considered to be less risky in the medical domain.

**Ablation experiment.** To investigate the influence of $\sigma_{\text{quality}}$ in the finetuning process, we conduct an ablation study based on the percentage of training data used to train HUMANPROXY and $\sigma_{\text{quality}}$. In our primary experiments, we take 10% of training examples to train the surrogate models. However, for the ablation, we finetune the baseline model $\sigma_{\text{quality}}$ with only 1% and 5% of the training examples. The results can be viewed in Fig. 7. While the surrogates trained with 5% of the data performed at par with surrogates trained with 10%, an improvement can still be observed when reducing the amount of training data further. In particular, macro $F_1$ (right) shows that even with 1% surrogate training data, SLOG still manages to outperform both baselines (dashed lines), while micro $F_1$ (left) is on-par with them. This indicates that even the less informed surrogates can help improving prediction quality for the rarer – but possibly significant for decision making – classes. In Table 11, we notice that SLOG trained with 10% of training examples outperforms the former two in both micro and macro metrics. Detailed results can be obtained in Table 12.

## 5  Related Work

**Aligning LLMs.** The standard approach for aligning large language models to human interests is reinforcement learning with human feedback (RLHF) (Ziegler et al., 2020; Ouyang et al., 2022). Several RLHF-based approaches that target medical tasks exist. Yunxiang et al. (2023) and Wang et al. (2023a) proposed medical chat models obtained by fine-tuning existing architectures, while Bazi et al. (2023) introduced a specially designed vision transformer. Seo et al. (2020) presented a method for improving the performance of an image caption generator with offline human feedback. SLOG can be viewed as a variant of RLHF that foregoes reinforcement learning in favor of a fully end-to-end fine-tuning strategy, for efficiency. It also differs in aim, in that it optimizes the model's guidance for *a specific human decision making task*, rather than for factuality and fairness in general (Ouyang et al., 2022).

**Pathological report generation.** Several approaches (Hou et al., 2021; Chen et al., 2020; Wang et al., 2022; 2023b) have been developed for machine-driven pathological report generation from chest X-ray images using the `Mimic-CXR-IV` (Johnson et al., 2019) and the Indiana University chest X-ray data sets (Demner-Fushman et al., 2016). (Lovelace & Mortazavi, 2020b) designed a model with a similar objective but they proposed to leverage the `CheXpert` dataset to enhance the coherence of their model. (Tanida et al., 2023) introduced a region-guided model to generate pathological reports, thus opening the window of interactive


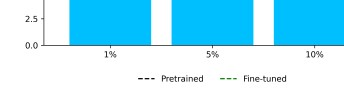

(a) Micro $F_1$ on the test data    (b) Macro $F_1$ on the test data

Figure 7: Performance of the finetuned models involving $\sigma_{\text{quality}}$ trained with varying percentage of train examples. The black and green lines are the $F_1$ scores obtained with the pretrained and finetuned models ($\lambda = 0$).

Table 8: Performance of `R2GenMamba` finetuned with SLOG. Results show per-class, macro and micro averaged precision, recall and $F_1$. Best $F_1$ results are boldfaced.

| PATHOLOGY | R2GenMamba | | | R2GenMamba (fine-tuned) | | | SLOG | | |
|---|---|---|---|---|---|---|---|---|---|
| | $Pr$ | $Rc$ | $F_1$ | $Pr$ | $Rc$ | $F_1$ | $Pr$ | $Rc$ | $F_1$ |
| No Finding | 32.97 | 82.81 | 47.16 | 37.36 | 69.23 | 48.53 | 42.02 | 60.18 | **49.49** |
| Enlarged Cardiomediastinum | 2.63 | 11.11 | **4.26** | 0.00 | 0.00 | 0.00 | 0.00 | 0.00 | 0.00 |
| Cardiomegaly | 15.32 | 9.04 | 11.37 | 19.10 | 27.13 | 22.42 | 19.38 | 43.09 | **26.73** |
| Lung Lesion | 0.00 | 0.00 | 0.00 | 0.00 | 0.00 | 0.00 | 0.00 | 0.00 | 0.00 |
| Lung Opacity | 33.33 | 1.71 | 3.26 | 29.49 | 5.62 | 9.45 | 24.42 | 10.27 | **14.46** |
| Edema | 62.75 | 10.29 | 17.68 | 67.86 | 12.22 | 20.71 | 50.43 | 18.65 | **27.23** |
| Consolidation | 8.33 | 2.82 | 4.21 | 9.09 | 1.41 | 2.44 | 5.48 | 5.63 | **5.56** |
| Pneumonia | 0.00 | 0.00 | 0.00 | 0.00 | 0.00 | 0.00 | 0.00 | 0.00 | 0.00 |
| Atelectasis | 25.62 | 13.72 | 17.87 | 23.83 | 22.57 | 23.18 | 22.13 | 36.73 | **27.62** |
| Pneumothorax | 0.00 | 0.00 | 0.00 | 15.38 | 13.33 | 14.29 | 21.05 | 13.33 | **16.33** |
| Pleural Effusion | 66.84 | 34.34 | 45.37 | 58.30 | 45.33 | 51.00 | 56.62 | 55.22 | **55.91** |
| Pleural Other | 0.00 | 0.00 | 0.00 | 0.00 | 0.00 | 0.00 | 0.00 | 0.00 | 0.00 |
| Fracture | 0.00 | 0.00 | 0.00 | 0.00 | 0.00 | 0.00 | 0.00 | 0.00 | 0.00 |
| Support Devices | 29.35 | 48.86 | 36.67 | 25.77 | 52.27 | 34.52 | 26.47 | 66.48 | **37.86** |
| MACRO | 19.80 | 15.34 | 13.42 | 20.44 | 17.79 | 16.18 | 19.14 | 22.11 | **18.66** |
| MICRO | 33.50 | 26.69 | 29.71 | 32.96 | 29.20 | 30.97 | 31.69 | 34.20 | **32.90** |

Table 9: Performance comparison between baseline (fine-tuned) and SLOG when a subset of samples was presented to a doctor. Results show per-class, macro and micro averaged precision, recall and $F_1$. Best $F_1$ results between baseline and SLOG are boldfaced.

| PATHOLOGY | R2GenCMN (Fine-tune) | | | | SLOG | | | |
|---|---|---|---|---|---|---|---|---|
| | $Pr$ | $Rc$ | $F_1$ | Bootstrap $F_1$ [95% CI] | $Pr$ | $Rc$ | $F_1$ | Bootstrap $F_1$ [95% CI] |
| No Finding | 23.53 | 44.44 | **30.77** | 29.60 [8.30, 53.30] | 8.33 | 11.11 | 9.52 | 9.70 [0.00, 28.60] |
| Enlarged Cardiomediastinum | 0.00 | 0.00 | 0.00 | 0.00 [0.00, 0.00] | 0.00 | 0.00 | 0.00 | 0.00 [0.00, 0.00] |
| Cardiomegaly | 23.81 | 50.00 | 32.26 | 31.40 [8.70, 51.90] | 32.00 | 80.00 | **45.71** | 45.20 [22.80, 63.20] |
| Lung Lesion | 0.00 | 0.00 | 0.00 | 0.00 [0.00, 0.00] | 7.14 | 33.33 | **11.76** | 11.40 [0.00, 33.40] |
| Lung Opacity | 66.67 | 17.39 | 27.59 | 26.60 [6.70, 47.60] | 50.00 | 30.43 | **37.84** | 36.90 [17.10, 57.10] |
| Edema | 20.00 | 28.57 | 23.53 | 21.60 [0.00, 47.60] | 25.00 | 57.14 | **34.78** | 34.20 [9.50, 57.10] |
| Consolidation | 33.33 | 33.33 | 33.33 | 27.70 [0.00, 80.00] | 37.50 | 100.0 | **54.55** | 52.70 [0.00, 85.70] |
| Pneumonia | 0.00 | 0.00 | 0.00 | 0.00 [0.00, 0.00] | 20.00 | 11.11 | **14.29** | 13.10 [0.00, 40.00] |
| Atelectasis | 27.27 | 60.00 | **37.50** | 35.80 [14.30, 56.20] | 18.52 | 50.00 | 27.03 | 26.80 [6.50, 46.20] |
| Pneumothorax | 50.00 | 50.00 | **50.00** | 40.50 [0.00, 100.00] | 16.67 | 50.00 | 25.00 | 20.90 [0.00, 66.70] |
| Pleural Effusion | 66.67 | 62.50 | **64.52** | 64.60 [40.00, 82.40] | 50.00 | 56.25 | 52.94 | 52.20 [30.80, 71.80] |
| Pleural Other | 0.00 | 0.00 | 0.00 | 0.00 [0.00, 0.00] | 0.00 | 0.00 | 0.00 | 0.00 [0.00, 0.00] |
| Fracture | 0.00 | 0.00 | 0.00 | 0.00 [0.00, 0.00] | 0.00 | 0.00 | 0.00 | 0.00 [0.00, 0.00] |
| Support Devices | 15.00 | 42.86 | 22.22 | 21.60 [0.00, 42.90] | 27.27 | 85.71 | **41.38** | 40.40 [18.20, 62.50] |
| MACRO | 23.31 | 27.79 | 22.98 | — | 20.89 | 40.36 | **25.34** | — |
| MICRO | 25.00 | 36.00 | 29.51 | — | 24.34 | 46.00 | **31.83** | — |

Table 10: Comparison of Raw Diagnostic Counts of the physician's annotations: Baseline vs. SLOG

| Pathology | Baseline | | | | SLOG | | | |
|---|---|---|---|---|---|---|---|---|
| | TP | FP | TN | FN | TP | FP | TN | FN |
| No Finding | 4 | 13 | 33 | 5 | 1 | 11 | 35 | 8 |
| Enlarged Cardiomediastinum | 0 | 17 | 38 | 0 | 0 | 14 | 41 | 0 |
| Cardiomegaly | 5 | 16 | 29 | 5 | 8 | 17 | 28 | 2 |
| Lung Lesion | 0 | 3 | 49 | 3 | 1 | 13 | 39 | 2 |
| Lung Opacity | 4 | 2 | 30 | 19 | 7 | 7 | 25 | 16 |
| Edema | 2 | 8 | 40 | 5 | 4 | 12 | 36 | 3 |
| Consolidation | 1 | 2 | 50 | 2 | 3 | 5 | 47 | 0 |
| Pneumonia | 0 | 3 | 43 | 9 | 1 | 4 | 42 | 8 |
| Atelectasis | 6 | 16 | 29 | 4 | 5 | 22 | 23 | 5 |
| Pneumothorax | 1 | 1 | 52 | 1 | 1 | 5 | 48 | 1 |
| Pleural Effusion | 10 | 5 | 34 | 6 | 9 | 9 | 30 | 7 |
| Pleural Other | 0 | 3 | 52 | 0 | 0 | 5 | 50 | 0 |
| Fracture | 0 | 2 | 52 | 1 | 0 | 3 | 51 | 1 |
| Support Devices | 3 | 17 | 31 | 4 | 6 | 16 | 32 | 1 |

Table 11: Ablation study on dataset size for SLOG.

| Type | Metric | slog 1% | slog 5% | slog 10% |
|---|---|---|---|---|
| **Macro** | Precision | 21.56 | 21.51 | 24.59 |
| | Recall | 23.15 | 23.54 | 23.31 |
| | $F_1$ | 19.72 | 20.12 | 20.25 |
| **Micro** | Precision | 30.06 | 30.02 | 29.98 |
| | Recall | 33.96 | 34.40 | 34.40 |
| | $F_1$ | 31.89 | 32.06 | 32.04 |

human-guided report generation. (Srivastav et al., 2024) used a large language model based on Vicuna-7B to generate radiology reports out of CXR images. A slightly different approach was used by (Woźnicki et al., 2024) where the authors used a large language model to extract the structured information out of the *findings* section of a report. However, these models are not concerned with optimizing the utility of the generated reports for the follow-up decision making. Our approach builds on top of these methods, enriching them with the ability to incorporate surrogate quality information (we use (Chen et al., 2020) in our evaluation, but any of these models can be adapted for SLOG).

**Other approaches.** LTG is related to *explain then predict* (ETP) (Camburu et al., 2018; Kumar & Talukdar, 2020; Zhao & Vydiswaran, 2021), a framework for building explainable (Guidotti et al., 2018) models in which a machine first outputs a full-fledged explanation – playing the role of "guidance" – and then derives a prediction from the explanation itself. In LTG, however, the prediction step is carried out by a human expert, and as such it is not differentiable. Also, ETP requires direct supervision on the explanations themselves, which is seldom available. In contrast, SLOG improves guidance quality using indirect scoring feedback, which is comparatively easier to acquire.

Finally, LTG is not restricted to textual guidance. One option is, for instance, to implement guidance in terms of explanations extracted from (or output by) an underlying image classifier to guide human decision making (Guidotti et al., 2018). From this perspective, LTG is tied to explanatory interactive learning (XIL) (Schramowski et al., 2020; Teso et al., 2023), in which the goal is to improve the quality of explanations output by a machine learning model by interactively acquiring corrections to the explanations themselves. The key difference is that LTG focuses on down-stream decision quality and SLOG supports textual guidance, while XIL aims at more generally improving explanation quality and implementations do no support textual explanations.

# 6 Discussion

One of the fundamental properties of SLOG is it works on the top of a pretrained model. Therefore, the performance of SLOG varies over the performance of the pretrained model. We also tried to experiment the performance of SLOG with respect to LLava model. However, at its primary inferential stage, the pretrained LLava model yielded sub-optimal results (13.66 % Micro $F_1$), possibly because the model is not particularly designed to perform complex medical tasks. Considering the models that have been tried in this study, the comparatively low performance of the llava model restricted us from assessing the performance of SLOG on the top of Llava as SLOG cannot dramatically improve the performance of the pretrained model. The core purpose of *learning to guide* is to provide guidance for the human decision maker. However, the performance of human may not be uniform and may depend on several other parameters and we reserved the analysis uniformity of impact of AI system on humans for future studies.

# 7 Conclusion

We introduced *learning to guide* as an alternative setup for high-stakes hybrid decision making that ensures the human expert is always in the loop, as well as SLOG, an end-to-end approach for turning pre-trained VLMs into high-quality textual guidance using human feedback. Our results suggest that SLOG is effective at steering VLMs towards generating more informative guidance, leading to improved accuracy in downstream decisions.

In follow-up work, we plan to extend SLOG by integrating ideas from active learning (Settles, 2012) to acquire the quality rankings, and to explore connections with explainable AI (Guidotti et al., 2018), explanatory interactive learning (Schramowski et al., 2020; Teso et al., 2023) and skeptical learning (Zeni et al., 2019) to facilitate the identification and correction of potential issues with the generated guidance.

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

# A   Appendix

Table 12: Ablation study of SLOG based on R2GenCMN with different proportions of labeled data (SLOG 1%, 5%, and 10%).

| PATHOLOGY | SLOG 1% | | | SLOG 5% | | | SLOG 10% | | |
|---|---|---|---|---|---|---|---|---|---|
| | $Pr$ | $Rc$ | $F_1$ | $Pr$ | $Rc$ | $F_1$ | $Pr$ | $Rc$ | $F_1$ |
| No Finding | 38.56 | 49.55 | 43.37 | 39.60 | 49.55 | 44.02 | 38.72 | 50.45 | 43.81 |
| Enlarged Cardiomediastinum | 1.27 | 5.56 | 2.06 | 1.39 | 5.56 | 2.22 | 0.00 | 0.00 | 0.00 |
| Cardiomegaly | 17.41 | 54.26 | 26.36 | 17.01 | 52.66 | 25.71 | 17.73 | 53.19 | 26.60 |
| Lung Lesion | 0.00 | 0.00 | 0.00 | 0.00 | 0.00 | 0.00 | 33.33 | 1.67 | 3.17 |
| Lung Opacity | 33.33 | 26.16 | 29.32 | 33.33 | 27.38 | 30.07 | 33.81 | 29.10 | 31.27 |
| Edema | 53.44 | 22.51 | 31.67 | 53.28 | 23.47 | 32.59 | 54.03 | 21.54 | 30.80 |
| Consolidation | 15.66 | 18.31 | 16.88 | 13.98 | 18.31 | 15.85 | 14.61 | 18.31 | 16.25 |
| Pneumonia | 14.29 | 1.31 | 2.40 | 8.33 | 1.31 | 2.26 | 14.29 | 1.96 | 3.45 |
| Atelectasis | 20.69 | 31.86 | 25.09 | 20.28 | 32.30 | 24.91 | 18.52 | 30.97 | 23.18 |
| Pneumothorax | 17.65 | 10.00 | 12.77 | 25.00 | 13.33 | 17.39 | 28.57 | 13.33 | 18.18 |
| Pleural Effusion | 64.22 | 40.93 | 50.00 | 64.02 | 42.03 | 50.75 | 64.44 | 39.84 | 49.24 |
| Pleural Other | 0.00 | 0.00 | 0.00 | 0.00 | 0.00 | 0.00 | 0.00 | 0.00 | 0.00 |
| Fracture | 0.00 | 0.00 | 0.00 | 0.00 | 0.00 | 0.00 | 0.00 | 0.00 | 0.00 |
| Support Devices | 25.28 | 63.64 | 36.19 | 24.94 | 63.64 | 35.84 | 26.19 | 65.91 | 37.48 |
| MACRO | 21.56 | 23.15 | 19.72 | 21.51 | 23.54 | 20.12 | 24.59 | 23.31 | 20.25 |
| MICRO | 30.06 | 33.96 | 31.89 | 30.02 | 34.40 | 32.06 | 29.98 | 34.40 | 32.04 |

---

**Algorithm 1:** Sampling with Minimum Class Coverage

**Input:** Dataset $D$ (rows = reports),
label set $\mathcal{C}$ (e.g., CheXpert classes),
target sample size $N$ (e.g., 30),
minimum positives per class $m$ (e.g., 2)
**Output:** Sampled subset $S \subseteq D$ of size $N$
$S \leftarrow \varnothing$ ;                                          // selected indices/reports
**foreach** $c \in \mathcal{C}$ **do**
  $P_c \leftarrow \{i \in D \mid \text{label}(i, c) = 1\}$;
  **if** $|P_c| \geq m$ **then**
    choose $m$ elements uniformly from $P_c$ without replacement and add to $S$;
  **else**
    add all of $P_c$ to $S$

**if** $|S| > N$ **then**
  uniformly sub-sample $S$ down to size $N$; **return** $S$;

$R \leftarrow D \setminus S$ ;                                        // remaining pool
**while** $|S| < N$ **and** $R \neq \varnothing$ **do**
  pick $x \in R$ uniformly at random; add $x$ to $S$; remove $x$ from $R$;
**return** $S$

---

Table 13: SLOG **vs R2Gen-Mamba (Pretrained & Fine-tuned) evaluated with ChexBert**

|  | R2Gen-Mamba (Pre) | | | R2Gen-Mamba (FT) | | | SLOG | | |
|---|---|---|---|---|---|---|---|---|---|
| PATHOLOGY | $Pr$ | $Rc$ | $F_1$ | $Pr$ | $Rc$ | $F_1$ | $Pr$ | $Rc$ | $F_1$ |
| Enlarged Card. | 7.04 | 3.36 | 4.55 | 6.25 | 2.68 | 3.76 | 15.00 | 6.04 | **8.61** |
| Cardiomegaly | 67.08 | 26.52 | 38.02 | 60.91 | 39.54 | 47.95 | 64.45 | 41.52 | **50.50** |
| Lung Opacity | 42.25 | 9.65 | 15.71 | 42.97 | 8.84 | 14.67 | 49.58 | 9.49 | **15.92** |
| Lung Lesion | 100.00 | 1.06 | **2.11** | 100.00 | 1.06 | **2.11** | 0.00 | 0.00 | 0.00 |
| Edema | 46.27 | 10.92 | 17.66 | 47.93 | 20.42 | **28.64** | 48.70 | 19.72 | 28.07 |
| Consolidation | 33.33 | 2.33 | 4.35 | 20.00 | 2.33 | 4.17 | 27.27 | 3.49 | **6.19** |
| Pneumonia | 25.00 | 4.00 | 6.90 | 50.00 | 8.00 | **13.79** | 50.00 | 8.00 | **13.79** |
| Atelectasis | 31.27 | 30.66 | 30.96 | 33.07 | 34.81 | **33.92** | 32.90 | 34.81 | 33.83 |
| Pneumothorax | 30.77 | 27.59 | **29.09** | 25.00 | 17.24 | 20.41 | 26.32 | 17.24 | 20.83 |
| Pleural Effusion | 75.93 | 40.94 | 53.20 | 67.61 | 53.24 | **59.57** | 66.02 | 53.02 | 58.81 |
| Pleural Other | 0.00 | 0.00 | 0.00 | 0.00 | 0.00 | 0.00 | 0.00 | 0.00 | 0.00 |
| Fracture | 0.00 | 0.00 | 0.00 | 0.00 | 0.00 | 0.00 | 0.00 | 0.00 | 0.00 |
| Support Devices | 74.75 | 41.76 | 53.58 | 75.63 | 54.58 | 63.40 | 75.99 | 56.23 | **64.63** |
| No Finding | 9.34 | **85.26** | 16.84 | 10.14 | 77.89 | 17.94 | 10.45 | 78.95 | **18.45** |
| MACRO | 38.79 | 20.29 | 19.50 | 38.54 | 22.90 | 22.17 | 33.33 | **23.46** | **22.83** |
| MICRO | 37.46 | 24.81 | 29.85 | 42.46 | 31.42 | 36.12 | **43.81** | **32.22** | **37.13** |

Table 14: SLOG **boosts quality of downstream decisions for `R2GenCMN-` evaluated with ChexBert**
Results show per-class, macro and micro averaged precision, recall and $F_1$. Best $F_1$ results are boldfaced.

|  | R2GenCMN | | | R2GenCMN (fine-tuned) | | | SLOG | | |
|---|---|---|---|---|---|---|---|---|---|
| PATHOLOGY | $Pr$ | $Rc$ | $F_1$ | $Pr$ | $Rc$ | $F_1$ | $Pr$ | $Rc$ | $F_1$ |
| Enlarged Cardiomediastinum | 12.68 | **6.21** | **8.33** | 8.43 | 4.83 | 6.14 | 8.75 | 4.83 | 6.22 |
| Cardiomegaly | 60.59 | 50.25 | 54.94 | 61.87 | **54.35** | 57.87 | **64.03** | 53.20 | **58.12** |
| Lung Opacity | 46.86 | **22.94** | 30.80 | 46.08 | 15.19 | 22.84 | **50.18** | 22.29 | **30.87** |
| Lung Lesion | 0.00 | 0.00 | 0.00 | 20.00 | 1.02 | 1.94 | **66.67** | **2.04** | **3.96** |
| Edema | 43.33 | 9.06 | 14.99 | 44.64 | 17.42 | 25.06 | **54.40** | **23.69** | **33.01** |
| Consolidation | 18.75 | 6.98 | 10.17 | 20.45 | 10.47 | 13.85 | **25.58** | **12.79** | **17.05** |
| Pneumonia | **24.00** | 8.00 | **12.00** | 18.75 | **8.00** | 11.21 | 13.89 | 6.67 | 9.01 |
| Atelectasis | 28.39 | 30.87 | 29.58 | **32.46** | 30.60 | **31.50** | 30.41 | **32.24** | 31.30 |
| Pneumothorax | 11.11 | 6.67 | 8.33 | **20.00** | 6.67 | 10.00 | **21.43** | **10.00** | **13.64** |
| Pleural Effusion | **80.90** | 31.93 | 45.79 | 72.82 | 33.26 | 45.66 | 74.22 | **37.03** | **49.41** |
| Pleural Other | 0.00 | 0.00 | 0.00 | 0.00 | 0.00 | 0.00 | 0.00 | 0.00 | 0.00 |
| Fracture | 0.00 | 0.00 | 0.00 | 0.00 | 0.00 | 0.00 | 0.00 | 0.00 | 0.00 |
| Support Devices | 77.87 | 53.58 | 63.48 | **80.31** | 47.16 | 59.42 | 80.71 | **54.50** | **65.06** |
| No Finding | 9.51 | 65.96 | 16.62 | 9.75 | 71.28 | 17.16 | **11.27** | **77.66** | **19.68** |
| MACRO | 29.57 | 20.89 | 21.07 | 31.11 | 21.45 | 21.62 | **35.82** | **24.07** | **24.09** |
| MICRO | 42.27 | 31.34 | 36.00 | 41.98 | 30.72 | 35.48 | **44.68** | **34.31** | **38.82** |

Table 15: SLOG **boosts quality of downstream decisions for R2Gen- evaluated with ChexBert** Results show per-class, macro and micro averaged precision, recall and $F_1$. Best $F_1$ results are boldfaced.

| | R2Gen | | | R2Gen (fine-tuned) | | | SLOG | | |
|---|---|---|---|---|---|---|---|---|---|
| PATHOLOGY | $Pr$ | $Rc$ | $F_1$ | $Pr$ | $Rc$ | $F_1$ | $Pr$ | $Rc$ | $F_1$ |
| Enlarged Cardiomediastinum | 8.82 | 2.07 | 3.35 | 7.69 | 2.07 | 3.26 | 7.14 | **3.45** | **4.65** |
| Cardiomegaly | 55.52 | **31.36** | **40.08** | 59.91 | 22.33 | 32.54 | **63.01** | 40.56 | **49.35** |
| Lung Opacity | **46.67** | **9.05** | **15.16** | 45.24 | 6.14 | 10.81 | 46.84 | 14.38 | **22.00** |
| Lung Lesion | 0.00 | 0.00 | 0.00 | 0.00 | 0.00 | 0.00 | 0.00 | 0.00 | 0.00 |
| Edema | 47.10 | 22.65 | 30.59 | **58.40** | **25.44** | **35.44** | 49.57 | 19.86 | 28.36 |
| Consolidation | 0.00 | 0.00 | 0.00 | 100.00 | 1.16 | 2.30 | 40.00 | 2.33 | **4.40** |
| Pneumonia | **50.00** | 6.67 | 11.76 | 23.08 | 4.00 | 6.82 | 37.50 | **8.00** | **13.19** |
| Atelectasis | **35.86** | **28.42** | **31.71** | 31.12 | 20.49 | 24.71 | 32.02 | 28.96 | 30.42 |
| Pneumothorax | 0.00 | 0.00 | 0.00 | 16.67 | 6.67 | 9.52 | **23.81** | **16.67** | **19.61** |
| Pleural Effusion | **78.67** | **36.81** | **50.15** | 73.98 | 32.15 | 44.82 | 71.55 | **36.81** | 48.61 |
| Pleural Other | 0.00 | 0.00 | 0.00 | 0.00 | 0.00 | 0.00 | 0.00 | 0.00 | 0.00 |
| Fracture | 0.00 | 0.00 | 0.00 | 0.00 | 0.00 | 0.00 | 0.00 | 0.00 | 0.00 |
| Support Devices | 77.95 | 46.06 | 57.90 | 77.23 | 42.94 | 55.19 | **79.94** | **50.46** | **61.87** |
| No Finding | 9.79 | **85.11** | **17.56** | 8.91 | **85.11** | 16.13 | **9.96** | 78.72 | **17.68** |
| MACRO | 29.31 | 19.16 | 18.45 | **35.87** | 17.75 | 17.25 | 32.95 | **21.44** | **21.44** |
| MICRO | 40.22 | 26.05 | 31.62 | 36.92 | 22.35 | 27.84 | **41.97** | **29.19** | **34.43** |

