# OpenReview forum: "Learning to Guide Human Decision Makers with Vision-Language Models"
_TMLR — Rejected by TMLR_

### Review · Reviewer_bacp · 2026-01-13

**Summary Of Contributions:**

The paper introduces a framework, namely SLOG, which is used to generate captions for given medical images based on pre-trained vision–language models. The generated captions can be used to support experts in decision-making. The framework is evaluated on both a synthetic dataset (in a controlled setting) and a real-world dataset. Experimental results show that the proposed framework improves the quality of downstream decision-making.

Strengths

S1. The approach is interesting, especially in scenarios where the vision–language model only provides guidance for decision-making.

S2. The experiments are conducted under multiple settings and on both synthetic and real-world datasets.

Weaknesses

W1. The quality of the generated text from the vision–language models is assessed using BLEU and BLEURT metrics. However, the paper does not discuss or address the hallucination issue of language models, nor how such hallucinations are mitigated. Even when BLEU/BLEURT scores are high, the generated text may still contain hallucinated content and thus be unreliable.

W2. The quality of the downstream decision-making is relatively low (F1 scores are below 50% in most cases shown in Tables 6, 7, and
8), which may limit the practical applicability of the proposed framework.

W3. Regarding the choice of vision–language models, it is unclear why the paper considers only a simple transformer-based model in the first experiment (Q1), given that this model is relatively basic and outdated. Additionally, the two selected models (R2Gen and its extension R2GenCMN) are also relatively old, having been introduced in 2020 and 2022, respectively.

W4. The hyperparameter settings are insufficiently described. For example, the paper does not explain how the number of training epochs, the parameter \alpha, or the value of k in the k-fold cross-validation procedure (Section 4.1) are chosen.

**Audience:**

Yes

**Audience Explanation:**

The study may be useful for researchers working on vision–language models and their real-world applications.

**Broader Impact Concerns:**

No further concerns are identified.

**Claims And Evidence:**

Yes

**Claims Explanation:**

The experimental results show that the proposed framework is promising, and they answered the given research questions.

**Requested Changes:**

See weaknesses.

---

### Review · Reviewer_u1k9 · 2026-02-14

**Summary Of Contributions:**

The paper introduces Learning to Guide (LTG), a novel framework for high-stakes decision-making where AI agents generate interpretable textual guidance to assist human experts rather than replacing them or deferring decisions entirely. SLOG (Surrogate-based Learning to Guide) has been proposed as a method that fine-tunes VLMs by using a surrogate model to estimate the quality of downstream decisions based on the generated text. The authors demonstrate the effectiveness of SLOG on both a synthetic dataset (CLEVR) and a real-world medical diagnosis task (MIMIC-CXR), showing that optimizing for decision utility yields more informative guidance than standard captioning objectives.

**Additional Comments:**

Overall, this is a well-written and truthful work, and I enjoyed reading the manuscript; however, I do recommend that the authors revise the human evaluation section to be reflected more clearly.

**Audience:**

Yes

**Audience Explanation:**

The paper touches on an important topic in medical AI, which is the hybrid-decision making problem, and the approach provided has technical weights.

**Broader Impact Concerns:**

The authors, for the human evaluation experiment, need to report the experimental methodology clearly, as mentioned above, since it can be misinterpreted by readers.

**Claims And Evidence:**

No

**Claims Explanation:**

Regarding the human evaluation in Section 4.2 and Table 8, I have significant concerns about the experimental design and the validity of the reported results. The pulmonologist’s absolute performance is implausibly low (F1 < 30%) for a specialist task, which seems inconsistent with the motivation for hybrid decision-making: the relevant baseline should be the clinician’s unaided performance, and the role of the AI is to improve human decision-making rather than outperform an unusually weak baseline. To clarify the rigor of this study, please address the following:

Was the experiment fully controlled? Specifically, did the physician evaluate the same cases using guidance from both models (paired design), or were the cases drawn from disjoint sets?

Was the experiment blinded? In particular, was the evaluator unaware of which model generated each piece of guidance?

Why is the expert’s performance so low? Does it reflect a deeper misalignment between the ground-truth labels and the task as presented to the clinician?

**Requested Changes:**

My requested changes are based on the concern I raised above:
1. Clarify and, if needed, redo the Human Evaluation with a controlled paired design. This should be explicit to say whether the physician evaluated the same cases under both model-guidance conditions. If not, rerun the human study so each case is evaluated twice (once with each model’s guidance), with randomized order, and report paired comparisons (including confidence intervals and a paired statistical test).

2. Please ensure that the physician does not know the source model, as it can cause some bias in their evaluation. (Or explicitly state in the body if the experiment has been done in such a way)

3. More importantly, include a proper clinician-only baseline and explain the low absolute performance. Report the clinician’s performance without AI guidance on the same sampled cases. In addition, provide an analysis explaining the low F1. If the low scores are driven by sampling or label/task misalignment, adjust the evaluation to better reflect a realistic clinical distribution and/or clarify the labeling/task definition, and report results under revised settings. This is important as the future reader can easily misinterpret the claims and the experiment supporting the claims.

---

> ### Author Response · Authors · 2026-03-01
>
> > **Was the experiment fully controlled? > Was the experiment blinded?**
>
> We are thankful for the constructive feedback. For both the baseline and SLOG, the clinician evaluated the cases with identical studies, ensuring a fair competition of the quality of guidances produced with the two models. We picked one unique study from the test data and provided the guidance generated with SLOG and its competitor. Thus, the clinician annotated the pathologies twice for the same studies.
>
> We confirm that the experiments were also absolutely blinded. The clinician was instructed to annotate 14 pathologies based on the guidance. However, during this process, the clinician had no scope to identify the source of the generated texts.
>
> > **Why is the expert’s performance so low? Does it reflect a deeper misalignment between the ground-truth labels and the task as presented to the clinician?**
>
> The primary reason for the expert’s performance being low is due to the unavailability of the chest x-ray images during the test. We purposefully did not provide the images along with the guidance because that would create a confounding factor during the post-hoc analysis, and thus make it difficult to assess the impact of SLOG. Our particular focus remains on the analysis that the texts generated with SLOG stand more beneficial than its competitors. In the process of preparing this rebuttal, we identified and corrected a technical inconsistency in the evaluation script. We have updated all the tables. We sincerely apologise for the inconvenience occurred from our side. However, the revised values neither essentially nullify the reviewer's comments nor discard the trend of the results previously reported.

---

### Review · Reviewer_NaKa · 2026-02-14

**Summary Of Contributions:**

This paper tackles an important problem and the core insight about automation bias and unassisted deferral is valuable. The LTG framework is conceptually sound and the motivation for keeping humans continuously in the loop is compelling for high-stakes domains.

**Additional Comments:**

The authors mention attempting to fine-tune pretrained state-of-the-art VLMs, specifically LLaVA 7B, but omitting these results because "fine-tuning failed to give sensible results, with the fine-tuned Llava performing much worse than our R2Gen model." This is precisely the kind of negative result that should be included in the paper rather than omitted. The failure to successfully apply SLOG to general-purpose VLMs while succeeding with the domain-specialized R2Gen raises critical questions about the method's scope and applicability. Does SLOG only work on models already adapted to the target domain? What specifically went wrong with LLaVA?

Including these negative results would significantly benefit the community by helping future researchers understand when SLOG is likely to succeed versus fail. The paper should report the LLaVA experiments in detail: what hyperparameters were tried, what interventions were attempted to fix the problems, what the failure mode looked like (poor text quality, poor decision quality, or both), and what hypotheses might explain why R2Gen succeeded where LLaVA failed. This could lead to valuable insights: perhaps SLOG requires a certain minimum level of domain adaptation in the base model, or perhaps it requires architectural features present in R2Gen but not LLaVA.

**Audience:**

Yes

**Audience Explanation:**

Beyond the conceptual framework, SLOG presents a methodological innovation in how to align language models to human preferences and downstream objectives. This is a very interesting work and as a researcher in AI for Healthcare, I think this paper makes great contributions to AI Alignment and AI Application domains.

The paper tackles a problem of substantial practical and societal importance: how to deploy AI in high-stakes domains while maintaining meaningful human oversight and trustworthiness. This aligns directly with TMLR's scope, which explicitly welcomes work on AI safety, robustness, fairness, and trustworthy ML systems.

**Broader Impact Concerns:**

More broadly, the paper needs substantive discussion of the theoretical and empirical conditions under which SLOG is expected to work. Under what circumstances should we expect fine-tuning with a surrogate model to improve actual human decision quality? The method assumes that the surrogate can accurately predict decision quality from image-text pairs, that the VLM can learn to optimize this surrogate through gradient descent, and that improvements on the surrogate correlate with improvements on real human decisions. Each of these assumptions could fail in various ways. The paper should also discuss what amount of training data the surrogate requires, since the ablation shows even 1% helps. Is there a point of diminishing returns, and how does this depend on task complexity or label dimensionality? A frank discussion of SLOG's limitations and boundary conditions would strengthen rather than weaken the contribution by setting appropriate expectations and guiding future research.

**Claims And Evidence:**

Yes

**Claims Explanation:**

With only 25 examples per method, these differences are likely not statistically significant. More importantly, the results are extremely mixed: SLOG helps on some pathologies but actively hurts on others. This suggests the method doesn't reliably improve human decision-making; it just shifts performance around unpredictably.

Also, the trade-offs aren't explained. Why does SLOG help on Atelectasis but hurt on Edema? Is this because the surrogate was trained on imbalanced data? Because the physician trusted guidance that was wrong? Understanding failure modes is critical. From the results, seems like SLOG helps on rare, serious conditions (good!) but hurts on common, easily-diagnosed ones (maybe acceptable?), this could still be valuable. But the author(s) need to frame it that way and weight the evaluation accordingly. What if you just showed the physician the actual ground-truth findings section? Would that outperform SLOG? This is the most important baseline the author(s) are missing.

Last but not least, statistical rigor must also be substantially improved. The paper should include formal significance testing using appropriate methods for paired predictions, such as McNemar's test or bootstrap confidence intervals for F1 differences. Without these statistical analyses, readers cannot determine whether the observed 3.73-point micro F1 improvement reflects a genuine benefit of SLOG or simply random variation across a small sample. The current presentation of raw numbers without uncertainty quantification or hypothesis testing falls below the standards expected for empirical ML research at a venue like TMLR.

**Requested Changes:**

It will be great to add critical analysis involves examining the guidance itself. For the cases where SLOG hurt performance, was the generated guidance factually incorrect, leading physicians astray? Was it too verbose, increasing cognitive load and decision time? Did it use imprecise medical terminology that physicians misinterpreted? A qualitative review session with the participating physician(s), examining specific failure cases and discussing what went wrong, would provide invaluable insights. Additionally, the paper should investigate whether the surrogate model's confidence in its quality estimates correlates with actual physician benefit: if the surrogate can identify when its own predictions are unreliable, this could enable selective deployment where guidance is only shown for cases where the surrogate is confident it will help.

---

> ### Author Response · Authors · 2026-03-01
>
> > **With only 25 examples per method, these differences are likely not statistically significant**
>
> We thank the reviwers for this insightful feedback. We conducted the experiment with the physician for additional 25 examples (we conducted with 30 in the first stage. The 25 was a typo and we rectified it), amounting to a total of 55 examples. In order to estimate the reliability, we added bootstrap CIs along with the evaluated efficacy scores. Our point estimates showcase an improvement on 7 out of 14 classes and SLOG performs worse on 4 out of 12 classes.
>
> > **Broader impact concerns**
>
> We Thank the reviewer for the insightful comments. We acknowledge the importance of this discussion and added a ‘Discussion’ section in the manuscript.
>
>
> we examined the pattern of error of the two models based on their conofusion matrix transitions (Baseline -> SLOG). We focused on the pathologies responsible for largest inflation in error, i.e. $\Delta FP + \Delta FN$.  In particular Lung lesion (+10 FP), Atelectasis (+6 TP, -1 TP) and Pleural Effusion (+4 FP, -1 TP). It is worth noticing that the dominant source of error is due to over predicting, rather than missing true pathologies.  Furthermore, rare instances of positive $\Delta$ in FN bolsters the fact that the lower performance of SLOG majorly stems from over detection, rather than missing actual cases – a phenomenon often considered to be less risky in the medical domain. We also included this discussion in the section 4.2 - Q4.
>
> ### Comparison of Raw Diagnostic Counts: Baseline vs. SLOG
>
> | Pathology | A TP | A FP | A TN | A FN | B TP | B FP | B TN | B FN |
> |------------|------|------|------|------|------|------|------|------|
> | No Finding | 4 | 13 | 33 | 5 | 1 | 11 | 35 | 8 |
> | Enlarged Cardiomediastinum | 0 | 17 | 38 | 0 | 0 | 14 | 41 | 0 |
> | Cardiomegaly | 5 | 16 | 29 | 5 | 8 | 17 | 28 | 2 |
> | Lung Lesion | 0 | 3 | 49 | 3 | 1 | 13 | 39 | 2 |
> | Lung Opacity | 4 | 2 | 30 | 19 | 7 | 7 | 25 | 16 |
> | Edema | 2 | 8 | 40 | 5 | 4 | 12 | 36 | 3 |
> | Consolidation | 1 | 2 | 50 | 2 | 3 | 5 | 47 | 0 |
> | Pneumonia | 0 | 3 | 43 | 9 | 1 | 4 | 42 | 8 |
> | Atelectasis | 6 | 16 | 29 | 4 | 5 | 22 | 23 | 5 |
> | Pneumothorax | 1 | 1 | 52 | 1 | 1 | 5 | 48 | 1 |
> | Pleural Effusion | 10 | 5 | 34 | 6 | 9 | 9 | 30 | 7 |
> | Pleural Other | 0 | 3 | 52 | 0 | 0 | 5 | 50 | 0 |
> | Fracture | 0 | 2 | 52 | 1 | 0 | 3 | 51 | 1 |
> | Support Devices | 3 | 17 | 31 | 4 | 6 | 16 | 32 | 1 |
>
> > **Statistical rigor**
>
> We calculated the Bootstrap CIs for both the competing cases. The mean F1 scores are now reported in Table 9, section 4.2. We also add the lower and upper bound of the CIs in the. To further investigate the reliability of these results, we calculated the bootstrap confidence intervals based on 1000 iterations and the mean $F_1$ scores negligibly deviate from our initial point estimates.
>
> | Pathology | Baseline $F_1$ | Baseline CI Mean $F_1$ | SLOG $F_1$ | SLOG Mean $F_1$ |
> | :--- | :---: | :---: | :---: | :---: |
> | No Finding | **30.77** | 29.60 | 9.52 | 9.70 |
> | Enlarged Cardiomediastinum | 0.00 | 0.00 | 0.00 | 0.00 |
> | Cardiomegaly | 32.26 | 31.40 | **45.71** | 45.20 |
> | Lung Lesion | 0.00 | 0.00 | **11.76** | 11.40 |
> | Lung Opacity | 27.59 | 26.60 | **37.84** | 36.90 |
> | Edema | 23.53 | 21.60 | **34.78** | 34.20 |
> | Consolidation | 33.33 | 27.70 | **54.55** | 52.70 |
> | Pneumonia | 0.00 | 0.00 | **14.29** | 13.10 |
> | Atelectasis | **37.50** | 35.80 | 27.03 | 26.80 |
> | Pneumothorax | **50.00** | 40.50 | 25.00 | 20.90 |
> | Pleural Effusion | **64.52** | 64.60 | 52.94 | 52.20 |
> | Pleural Other | 0.00 | 0.00 | 0.00 | 0.00 |
> | Fracture | 0.00 | 0.00 | 0.00 | 0.00 |
> | Support Devices | 22.22 | 21.60 | **41.38** | 40.40 |
> | **MACRO** | 22.98 | — | **25.34** | — |
> | **MICRO** | 29.51 | — | **31.83** | — |

---

### Decision · Action_Editor_ycVZ · 2026-04-26

**Recommendation:** Reject

**Audience:**

Yes

**Audience Explanation:**

The submission focused on an important problem in the community of VLM.

**Claims And Evidence:**

No

**Claims Explanation:**

Reviewers raised concerns about the validity of the human evaluation, noting that the experimental design is unrealistic, the absolute performance is implausibly low, the sample size is small, and the results are highly mixed across pathologies without adequate explanation; these concerns are only partially addressed, as the authors added bootstrap confidence intervals but did not redesign the experiment to reflect a realistic clinical setting. Reviewers also raised concerns about the absence of a proper clinician-only baseline, the lack of statistical rigor for distinguishing genuine benefit from random variation, and the omission of negative LLaVA fine-tuning results that directly bear on the method's scope; these concerns remain largely unresolved after rebuttal. Additional concerns about hallucination in generated guidance, outdated backbone models, and insufficient hyperparameter reporting were raised and only partially addressed. In its current state, the paper's core claim of improving human decision-making in high-stakes medical settings is not yet backed by sufficiently rigorous and convincing evidence.